# Contact-dependent killing by *Caulobacter crescentus* via cell surface-associated, glycine zipper proteins

Leonor García-Bayona[1,2,3], Monica S Guo[1], Michael T Laub[1,2]*

[1]Department of Biology, Massachusetts Institute of Technology, Cambridge, United States; [2]Howard Hughes Medical Institute, Massachusetts Institute of Technology, Cambridge, United States; [3]Graduate Program in Microbiology, Massachusetts Institute of Technology, Cambridge, United States

**Abstract** Most bacteria are in fierce competition with other species for limited nutrients. Some bacteria can kill nearby cells by secreting bacteriocins, a diverse group of proteinaceous antimicrobials. However, bacteriocins are typically freely diffusible, and so of little value to planktonic cells in aqueous environments. Here, we identify an atypical two-protein bacteriocin in the α-proteobacterium *Caulobacter crescentus* that is retained on the surface of producer cells where it mediates cell contact-dependent killing. The bacteriocin-like proteins CdzC and CdzD harbor glycine-zipper motifs, often found in amyloids, and CdzC forms large, insoluble aggregates on the surface of producer cells. These aggregates can drive contact-dependent killing of other organisms, or *Caulobacter* cells not producing the CdzI immunity protein. The Cdz system uses a type I secretion system and is unrelated to previously described contact-dependent inhibition systems. However, Cdz-like systems are found in many bacteria, suggesting that this form of contact-dependent inhibition is common.

*For correspondence: laub@mit.edu

## Introduction

To survive within complex microbial communities such as those found in the guts of animals or in soil, bacteria have evolved, and now rely on, a sophisticated array of strategies that allow them to compete for a limited set of resources. This constant battle for space and nutrients often involves the secretion of diffusible antimicrobials, including small-molecule antibiotics and bacteriocins. The secretion of these toxic compounds typically provides a fitness advantage to the producing cell by inhibiting the growth of competing cells and, in some cases, by lysing these competitors to liberate nutrients (*Cornforth and Foster, 2013*; *Hibbing et al., 2010*). The production of many antimicrobials is induced by stress signals triggered by crowding or the low availability of specific nutrients (*Rebuffat, 2011*).

Bacteriocins are ribosomally-synthesized proteinaceous toxins that are also sometimes post-translationally modified (*Cotter et al., 2013*; *James et al., 1992*; *Nissen-Meyer et al., 2011*; *Riley and Wertz, 2002*). Most small bacteriocins are secreted into the environment through a type I secretion system (*De Kwaadsteniet et al., 2006*; *Rebuffat, 2011*). Bacteriocins can be encoded on plasmids or in chromosomal gene clusters that generally contain all of the genes necessary for their synthesis, modification, and secretion, along with an immunity gene that protects the producer cell from self-intoxication. The inhibitory activity of bacteriocins can be broad- or narrow-spectrum, often determined by the nature of their cellular targets or the receptor proteins on target cells that mediate uptake. Diverse cellular targets have been described for bacteriocins. However, many insert into the membranes of target cells, either alone or by associating with integral membrane proteins,

producing pores that alter cytosolic membrane permeability, causing the leakage of cellular contents, loss of membrane potential, and eventual cell death (*Cotter et al., 2013*; *Rebuffat, 2011*; *Vassiliadis et al., 2011*).

Producing diffusible, secreted bacteriocins may not be an efficient competitive strategy for many bacteria, particularly in certain growth conditions. For instance, *Caulobacter crescentus* is an α-proteobacterium that thrives in nutrient-poor aquatic conditions (*Poindexter, 1981*) where a secreted bacteriocin would be quickly washed away or diluted, making it ineffective in killing neighboring cells (*Aguirre-von-Wobeser et al., 2015*). Additionally, the production of a secreted toxin, or any 'public good', also renders a population of cells sensitive to the proliferation of so-called cheaters that do not pay the energetic cost of producing the toxin, but benefit from its production by others (*Riley and Gordon, 1999*; *Travisano and Velicer, 2004*). To circumvent these limitations of secreted toxins, some bacteria have evolved killing systems that require direct contact between a producer and a target cell. This includes the contact-dependent growth inhibition systems found in many Gram-negative pathogens in which a CdiA toxin is anchored to the outer membrane via a type V secretion mechanism, and then delivered directly to a target cell (*Aoki et al., 2005*, *2010*). Similarly, type VI and VII secretion systems are often used to deliver toxins to direct neighbors (*Cao et al., 2016*; *Hood et al., 2010*; *Russell et al., 2014*). Homologs of these proximity- or contact-dependent inhibition systems are absent from the *C. crescentus* genome (*Marks et al., 2010*). In fact, aside from an ability to adhere together and form biofilms, no social behavior or cell-cell interaction system such as quorum-sensing has been previously described for *Caulobacter*.

Here, we describe a novel, atypical bacteriocin system in *C. crescentus,* now called the contact-dependent inhibition by glycine zipper proteins (Cdz) system, that enables producing cells to kill other cells in a contact-dependent manner. The Cdz system bears some genetic similarity to the small unmodified two-peptide bacteriocins (class IIb) from Gram-positive bacteria. However, in surprising contrast, the *C. crescentus* Cdz system yields no inhibitory activity in culture supernatant and the two small proteins CdzC and CdzD are only found at very low concentrations in the supernatant. Instead, the CdzC/D proteins remain almost exclusively cell-associated, with the CdzC protein forming large heat- and SDS-stable aggregates via a glycine-zipper repeat structure often seen in amyloid proteins. Together, the surface-associated proteins CdzC and CdzD enable producer cells to kill neighboring cells through direct contact. The Cdz system is massively induced upon entry to stationary phase and drives a nearly complete killing of any neighboring cells lacking the CdzI immunity protein. A computational search identified homologous, but as yet uncharacterized, systems in a range of species, including several clinically relevant pathogens. Our results suggest that contact-dependent inhibition may be more widespread than previously appreciated and has arisen through the modification of systems traditionally thought to produce diffusible toxins.

## Results

### A *C. crescentus* bacteriocin gene cluster is induced in stationary phase

We hypothesized that, if they exist, *Caulobacter* genes involved in intercellular competition would be upregulated as cells enter stationary phase when cell density increases and nutrients become scarce. Thus, we used RNA-Seq to compare global patterns of gene expression in a culture of wild-type *Caulobacter crescentus* (CB15N) grown to mid-exponential and early stationary phase (*Figure 1A*). One cluster of genes (CC0681-CC0684 in CB15; CCNA_03931-CCNA_00720 in CB15N) was very highly upregulated, with two of the genes, CCNA_03932 and CCNA_03933, increasing nearly 50-fold in stationary phase relative to exponential phase, reaching levels higher than all other genes in stationary phase.

Careful inspection of this region of the genome revealed five open reading frames in two divergently transcribed operons (*Figure 1B*). We named these genes *cdzABCD* and *cdzI* for contact-dependent inhibition by glycine zipper proteins based on results below. The genes *cdzA* and *cdzB* are homologous to canonical type I secretion systems. The genes *cdzC* and *cdzD* are predicted to encode small (86 and 84 amino acids, respectively) proteins with weak similarity to bacteriocins (using the Gene Function Identification Tool in KEGG) and N-terminal regions resembling the signal peptides of secreted RTX proteins (*Kanonenberg et al., 2013*). These analyses suggested that *cdzC* and *cdzD* may represent a previously uncharacterized two-protein bacteriocin. The *cdzI* gene had no

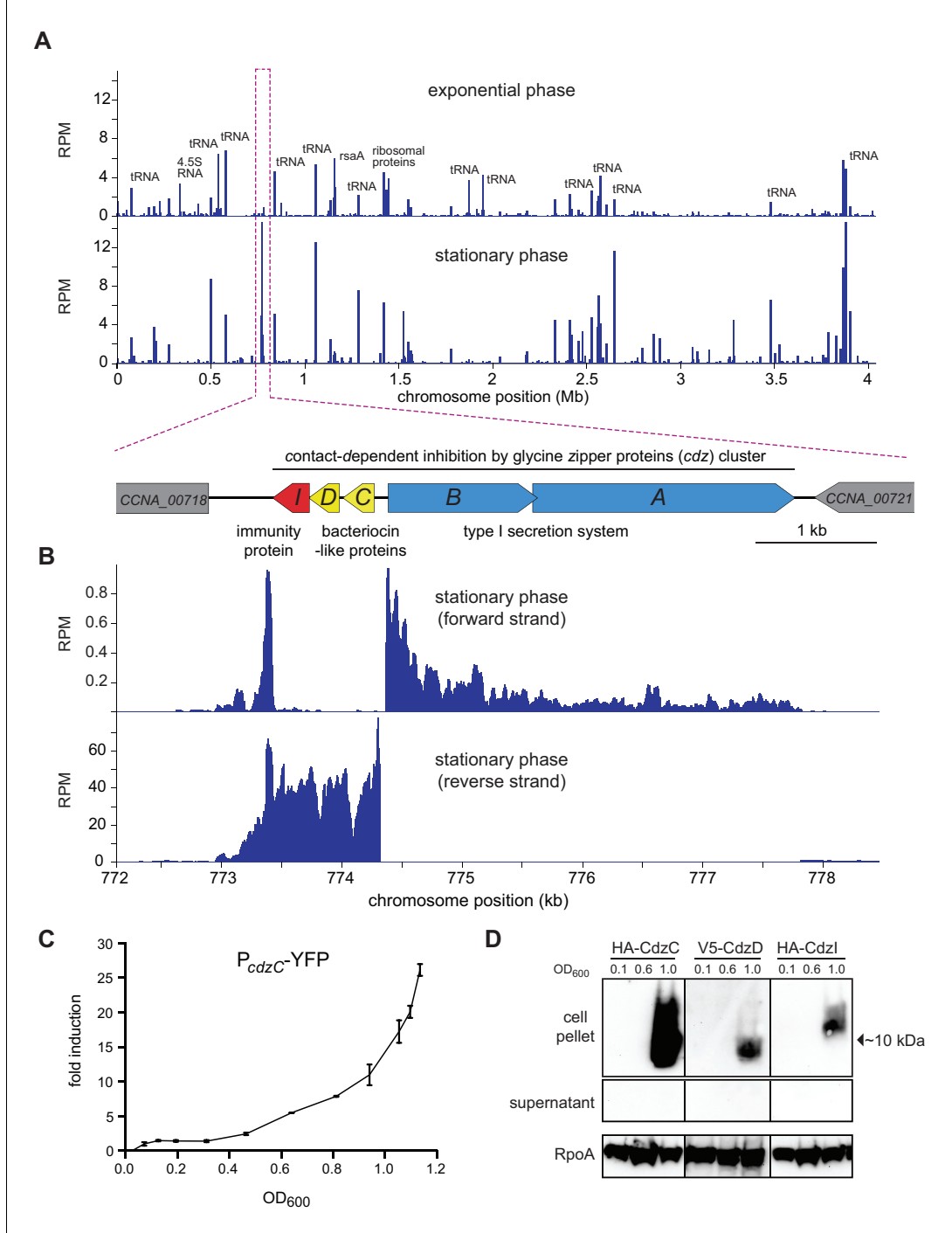

**Figure 1.** The *cdz* gene cluster is very highly expressed in stationary phase. (**A**) Genome-wide RNA-seq profiles of *C. crescentus* CB15N grown in rich medium to exponential (top) or stationary (bottom) phase. Expression levels (RPM) for both strands combined are plotted as a function of genomic position. A schematic of the *cdz* cluster is shown beneath the RNA-seq profiles. (**B**) RNA-seq profile of the *cdz* gene cluster in stationary phase. Each strand is shown separately. (**C**) Activity of a *cdzC* transcriptional reporter (YFP) as a function of culture density (for the corresponding growth curve, see *Figure 1—figure supplement 1*). Induction is reported relative to the level when $OD_{600}$ = 0.025. Data points are mean of three independent cultures; error bars indicate standard deviation (S.D.). (**D**) Western blot analysis of epitope-tagged CdzC, CdzD, and CdzI in the cell pellets of supernatant taken from cultures at the $OD_{600}$ values indicated. RpoA is a loading control.

The following figure supplement is available for figure 1:

**Figure supplement 1.** Growth curve for cells expressing a $P_{cdzC}$–YFP reporter; see Fig.

obvious sequence homologs, but was predicted by TM-HMM to harbor three transmembrane domains, and could represent an immunity gene that prevents self-intoxication by the bacteriocins (see below).

To confirm induction of the *cdz* genes upon entry to stationary phase, we generated a reporter construct in which the predicted promoter region of cdzC was fused to YFP. Cells were grown in a rich medium from early exponential phase (OD$_{600}$ ~ 0.05) into stationary phase (OD$_{600}$ ~ 1.2) (*Figure 1—figure supplement 1*). YFP levels increased dramatically as cells exited exponential phase, exhibiting ~10 fold induction once the culture reached an OD$_{600}$ of 1.0 and ~30 fold after reaching an OD$_{600}$ of 1.2 (*Figure 1C*). Expression of the *cdz* genes may also be regulated by post-transcriptional mechanisms as we noted in our RNA-Seq data that an antisense RNA overlapping the 3' end of the *cdzCDI* operon was expressed during exponential phase, but significantly less so during stationary phase.

To further examine *cdz* induction, we generated strains in which CdzC or CdzD harbored an epitope tag, HA or V5, respectively, immediately following the putative secretion signal; CdzI was tagged with HA at the N-terminus. Western blots of cell extracts indicated that none of these proteins (CdzC, CdzD, or CdzI) were detectable in cultures grown to an OD$_{600}$ of 0.1 or 0.6, but each was easily detected once cells reached an OD$_{600}$ of ~1.0 (*Figure 1D*), confirming their stationary phase-dependent induction. Surprisingly, the epitope-tagged CdzC and CdzD were detected only in cell pellets, not the cell-free supernatants, suggesting that they are not conventional, secreted bacteriocins and instead remain cell-associated, a possibility explored further below.

## Cells harboring the *cdz* gene cluster can kill cells lacking it

We hypothesized that CdzC and CdzD form a two-peptide toxin that is secreted via the type I secretion system CdzAB, with CdzI providing immunity against CdzC/D. To test this model and to genetically dissect the *cdz* system, we built an 'indicator' strain in which *cdzABCDI* was deleted. We then mixed this Δ*cdzABCDI* indicator with a 'producer', the wild-type *C. crescentus*, at a ratio of 1:1 and co-cultured the strains from early exponential phase into stationary phase. To quantify the presence of each strain, we inserted a gentamicin-resistance cassette into the wild-type producer and a tetracycline resistance cassette into the Δ*cdzABCDI* indicator; the colony-forming units of each strain were then assessed at various time points during co-culturing (*Figure 2A*).

The indicator and producer strains maintained a ratio of ~1:1 throughout exponential phase, up to a cell density of ~3 × 10$^9$ cells/mL (OD$_{600}$ ~0.9), at which point the Δ*cdzABCDI* strain began to lose viability, dropping nearly four orders of magnitude within ~7 hr. A similar result was obtained for strains in which the antibiotic resistance cassettes were swapped (*Figure 2—figure supplement 1A*). As a control, we verified that the Δ*cdzABCDI* indicator strain maintained viability and a 1:1 ratio when co-cultured with another Δ*cdzABCDI* strain harboring a different antibiotic resistance marker (*Figure 2—figure supplement 1B*). Similarly, two wild-type producer strains with different antibiotic resistance markers were maintained at a ratio of ~1:1, even in stationary phase (*Figure 2—figure supplement 1C*). Finally, we found that a Δ*cdzCDI* indicator strain, grown separately and only to exponential phase, was still killed when mixed 1:1 with a producer strain that had been grown separately to stationary phase (*Figure 2—figure supplement 1D*), indicating that the producer gains the ability to kill in stationary phase and susceptible cells need not be in stationary phase.

## CdzC, CdzD, and a type I secretion system are required for killing

To further dissect the requirements for Cdz-based killing we generated a series of producer strains and competed each against a Δ*cdzABCDI* indicator, mixing cultures at a 1:1 ratio upon entry to stationary phase (*Figure 2B*). The competitive index was then calculated as the ratio of indicator to producer at the final time point relative to the same ratio at the initial time point. Producer strains lacking *cdzC*, *cdzD*, *cdzCD*, or *cdzCDI* each failed to kill the indicator (*Figure 2C*), with competitive indices of ~1 for the indicator in each case, as also seen for a producer lacking the entire *cdz* cluster. For the Δ*cdzC* and Δ*cdzD* producers, full killing activity was restored when a copy of *cdzC* or *cdzD*, respectively, was provided on a plasmid (*Figure 2—figure supplement 1E*). Similarly, only a construct harboring both *cdzC* and *cdzD* complemented a Δ*cdzCD* strain (*Figure 2D*). Together, these results indicate that both bacteriocin-like proteins, CdzC and CdzD, are required for killing.

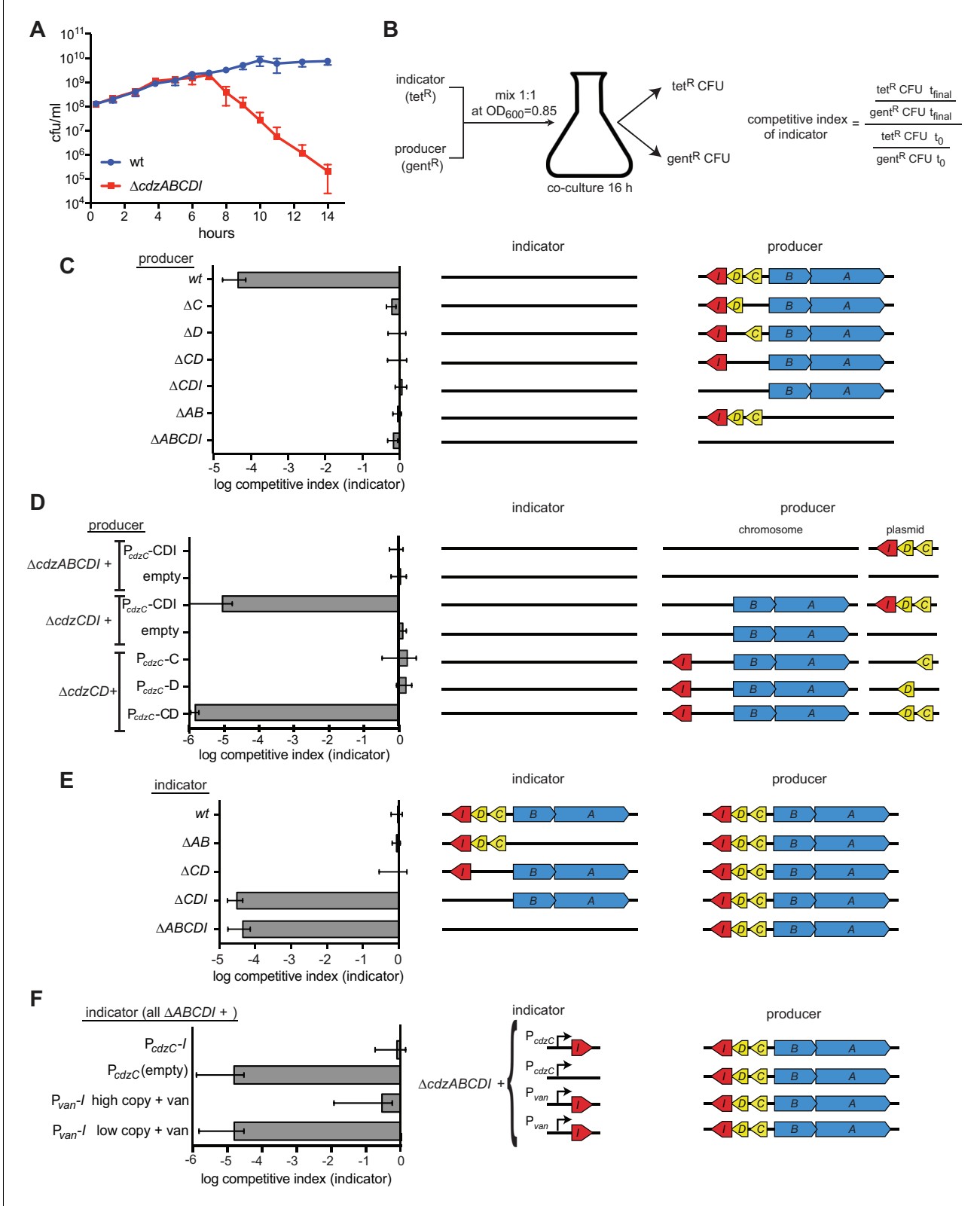

**Figure 2.** The *cdz* gene cluster encodes a two-peptide bacteriocin, secreted via a type I secretion system, and a small transmembrane immunity protein. (**A**) Co-culture competition of wild-type *C. crescentus* and Δ*cdzABCDI* in rich medium. Strains were mixed 1:1 in exponential phase and co-cultured, with CFUs of each strain measured up to 14 hr. (**B**) Experimental set-up for co-culture competitions in stationary phase and overview of competitive index calculation. (**C–F**) Competitions between the indicator and producer strains with the genotypes indicated schematically. Graphs show

*Figure 2 continued on next page*

*Figure 2 continued*

the competitive index of the indicator strain in each experiment. In each panel, data points represent the mean of at least three independent experiments; error bars indicate S.D. Induction of the P$_{van}$ promoter by vanillate is indicated by '+ van'.

The following figure supplement is available for figure 2:

**Figure supplement 1.** Control experiments for the genetic dissection of the Cdzc system.

The putative secretion system formed by CdzA and CdzB was also required for a producer strain to kill a Δ*cdzABCDI* indicator (*Figure 2C–D*). Canonical type I secretion systems comprise an ATP-binding cassette that resides in the inner membrane (CdzA), an adaptor protein in the periplasm (CdzB), and a TolC-like outer membrane protein (*Kanonenberg et al., 2013*). As a homolog of the latter is absent from the *cdz* cluster, we suspect killing also requires one of the two TolC homologs in the *C. crescentus* genome that likely associate with many different type I secretion systems.

Bacteriocin gene clusters often contain genes involved in post-translational modification of the toxic protein precursors (*James et al., 1992*; *Nissen-Meyer et al., 2011*). To more precisely define the bounds of the *cdz* genomic cluster, we made larger deletions, spanning upstream genes of unknown function, including a predicted membrane-associated CaaX motif protease, often present in bacteriocin clusters. However, the killing activity of strains carrying these larger deletions could be fully complemented by introducing only *cdzABCDI* on a plasmid (*Figure 2—figure supplement 1F*) indicating that (i) the neighboring genes are not part of the *cdz* system and (ii) CdzC and CdzD are likely not post-translationally modified, although we cannot fully rule out the possibility of modification by genes encoded elsewhere in the *Caulobacter* genome.

## Immunity to *cdz* is conferred by the small transmembrane protein CdzI

To identify the putative immunity gene, which would protect a producer strain from self-intoxication, we made several indicator strains with deletions spanning different regions of *cdz* and tested each for inhibition by a wild-type producer (*Figure 2E*). Our results indicated that *cdzI* was necessary for immunity, because any deletion including this gene led to killing by the producer, whereas the other *cdz* genes were dispensable for immunity. We could not generate a strain lacking only *cdzI* as this gene is essential for viability, consistent with it providing immunity to CdzC-CdzD. We also tested a Δ*cdzABCDI* strain producing only *cdzI* in trans (*Figure 2F*). Full resistance to a wild-type producer was seen if *cdzI* was driven by its native promoter (the region upstream of *cdzC*), or if *cdzI* was driven by a vanillate-inducible promoter on a high-copy vector with cells grown in the presence of vanillate. Taken together, these results indicate that CdzI is critical for immunity to CdzC-CdzD.

## CdzC and CdzD promote lysis of target cells

The co-culture competition experiments shown in *Figure 2* are consistent with the Cdz gene cluster encoding a system that can kill cells lacking the CdzI immunity protein. To more directly assess killing by CdzC/D-producing strains and to gain insight into the mechanism(s) of killing, we used fluorescence microscopy to examine co-cultured cells. We engineered a wild-type producer strain to synthesize CFP and a Δ*cdzCDI* indicator to synthesize YFP. These strains were then mixed at a 1:1 ratio in early stationary phase and co-cultured. At various time points after mixing, we imaged the co-cultured cells in both the YFP and CFP channels. We also stained cells at each time point with propidium iodide (PI) to label in red cells with compromised membrane integrity that are likely dead.

After 20 hr of co-culturing, >95% of the Δ*cdzCDI* indicator cells were stained by PI and thus were likely dead. In contrast, only ~10% of wild-type producer cells were stained by PI, a frequency comparable to that seen when CFP-labeled WT cells were competed against YFP-labeled WT cells (*Figure 3A–B*). After 20 hr, nearly all PI-positive indicator cells still had a typical, crescent-shaped morphology and were relatively dark in phase contrast. However, by 44 hr, 30% of PI-positive cells were no longer homogeneously dark in phase contrast, and 13% had become round, a signature of cell lysis. By 71 hr, 70% of the indicator population was comprised of round, PI-positive cells. These results demonstrated that Cdz producers do, in fact, kill cells lacking CdzI, leading to a loss of membrane integrity and, ultimately, cell death. We verified that expression of the fluorophores did not

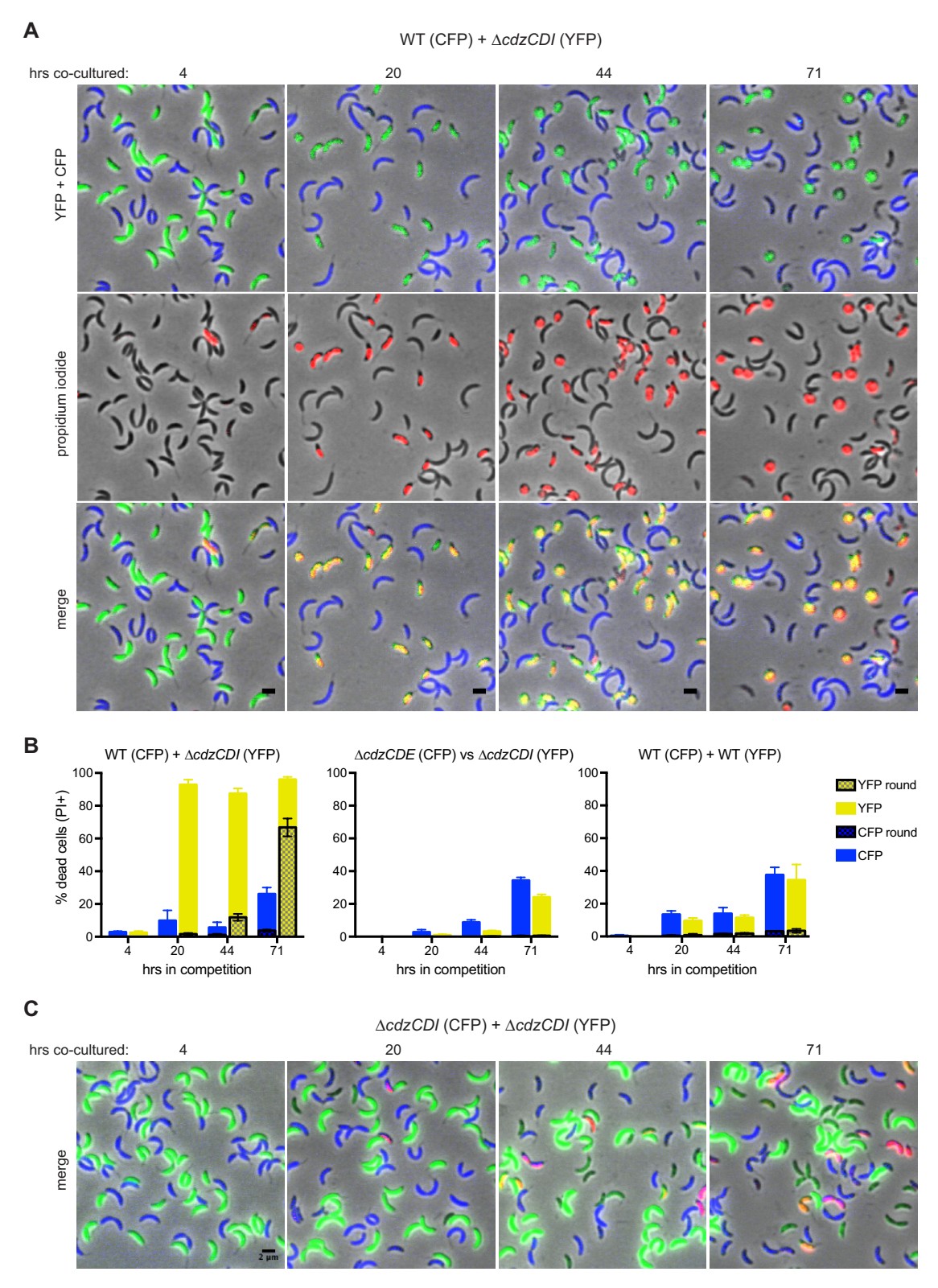

**Figure 3.** CdzC and CdzD kill an indicator strain in co-culture. (**A**) Microscopy analysis of aliquots taken from co-culture competitions of a wild-type producer synthesizing CFP (false colored blue) and a Δ*cdzCDI* indicator synthesizing YFP (false colored green), and stained with propidium iodide (red) to assess cell permeability. Fields of cells at the time points indicated were imaged by phase and epi-fluorescence microscopy. Phase images were overlaid with the corresponding YFP+CFP images (top), the propidium iodide image (middle), or both (bottom). Cells staining red and green are shown

*Figure 3 continued on next page*

*Figure 3 continued*

in yellow in the merged image (bottom). (B) Quantification of dead, propidium iodide-stained cells. At least 250 cells in three separate fields of view were counted for each time point of each competition; results were averaged, with error bars indicating S.D. (C) Same as panel A but for a co-culture competition of two indicator strains, synthesizing CFP or YFP.

The following figure supplement is available for figure 3:

**Figure supplement 1.** Control experiments for the microscopy-based analysis of Cdz-dependent killing.

affect the outcome of the competition by swapping CFP and YFP between producer and indicator (*Figure 3—figure supplement 1A*). As a control, we also verified that a co-culture of two indicator strains expressing different fluorophores did not lead to the same pervasive cell death (*Figure 3B–C*).

As noted, we also competed two wild-type producer strains that synthesize different fluorophores. We found that ~10–20% of cells from each population were PI-positive by ~20 hr, with some round cells appearing by day three (*Figure 3—figure supplement 1B*). The percentage of PI-positive cells was consistently higher at each time point relative to the experiment competing two Δ*cdzCDI* indicator strains. These results suggest that producer cells harboring CdzI may still be subject to occasional CdzC/D-mediated death, possibly due to variability in the timing or extent of CdzI production in individual cells.

## Cdz-based killing requires physical contact between producer and indicator cells

Although similar in some ways to previously described two-peptide bacteriocins (*Nissen-Meyer et al., 2011*), we suspected that CdzC/D were not secreted, diffusible proteins as both were detected by Western blot analysis in cell pellets, rather than in the culture supernatant (*Figure 1D*). Additionally, we found that the supernatant from a stationary phase culture of a producer strain did not kill a Δ*cdzABCDI* indicator strain (*Figure 4A*). We also tried concentrating the supernatant from a culture of producer cells using trichloroacetic acid (TCA) or ammonimum sulfate precipitation, or by using a centrifugal filter, but did not see any killing (*Figure 4A*), even though CdzC could now be detected in the concentrated supernatant (*Figure 4—figure supplement 1A*). These results, along with our initial immunoblots (*Figure 1D*), strongly suggested that CdzC and CdzD typically remain cell associated and may mediate contact- or proximity-dependent killing.

To directly assess whether CdzC/D-based killing requires cell-to-cell contact, we co-cultured strains using a split-well arrangement in which producer and indicator strains are separated by a porous membrane (*Figure 4B*). In this setup, any molecules secreted into the growth medium by the producer should be able to target the indicator, unless they form structures larger than the pore size or adhere to the membrane. Strikingly, killing of the indicator occurred when the membrane pores were 8 µm in diameter and able to allow the passage of whole cells, but not when the pores were 0.4 µm, which restricts the exchange of cells.

Our split-well experiments strongly suggested that CdzC/D-based killing requires direct cell contact. On their own, these experiments do not rule out the possibility that the toxins form large aggregates that are released from cells, but that cannot pass through the 0.4 µm filters used. However, neither CdzC nor CdzD was detected in culture supernatant (*Figure 1D*) and culture supernatant exhibited no killing activity, even if concentrated 2000X (*Figure 4A*). The preparation of such cell-free supernatants required the centrifugation and pelleting of cells, which could, in principle, also have pelleted large aggregates of toxins that had been released from cells to mediate non-contact-dependent killing. To rule out this possibility, we first killed stationary phase producer cells by boiling for 5 min before adding them, without a centrifugation step, to indicator cells. If large toxin aggregates were present in the supernatant, the indicator should still be killed, but it was not (*Figure 4A*). Similarly, we pre-treated stationary phase gentamicin-sensitive producer cells with gentamicin for 4 hr before adding them to gentamicin-resistant indicator cells, again without centrifugation. As with boiled producer cells, no killing was observed. Taken all together, these results support the conclusion that CdzC/D mediate contact-dependent killing.

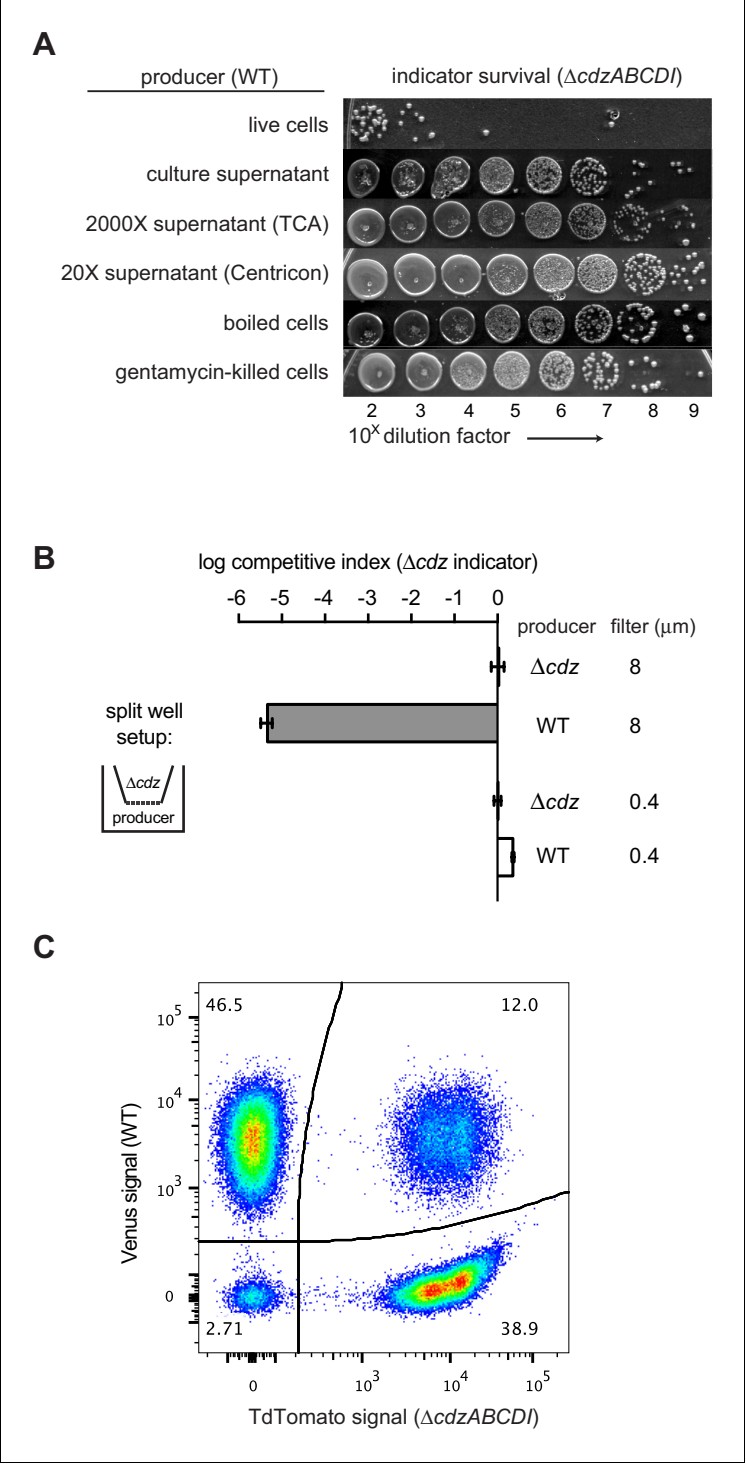

**Figure 4.** Cdz-mediated killing requires cell-cell contact between a producer and an indicator. (**A**) Survival of a Δ*cdzABCDI* indicator when treated for 16 hr with live producer cells, producer culture supernatant, producer culture supernatant concentrated 2000X by trichloroacetic acid precipitation, producer culture supernatant concentrated 20X by centrifugation, producer cells that were boiled for 5 min, or producer cells treated with gentamicin. (**B**) Competitive index of Δ*cdzABCDI* indicator grown in co-culture with a wild-type producer or Δ*cdzABCDI* mock-producer using the split-well set-up shown (left). The membrane separating the two compartments had a pore size of 8 or 0.4 μm, as indicated. (**C**) Flow cytometry analysis of a 1:1 mixture in stationary phase of wt producer cells expressing Venus and Δ*cdzABCDI* indicator cells expressing TdTomato.

*Figure 4 continued on next page*

*Figure 4 continued*
The following figure supplement is available for figure 4:
**Figure supplement 1.** Control experiments for the analysis of contact-dependent inhibition.

In some contact-dependent systems, like that mediated by type VI secretion systems, killing occurs primarily on solid substrates, not in culture (*Russell et al., 2014b*). However, the Cdz system appeared to drive cell contact-dependent killing even in culture, implying that cells must adhere together at some frequency. The *C. crescentus* strains we used were derived from the strain CB15N, which lacks the adhesive holdfast present in some *Caulobacter* isolates. To test whether the strains we used form stable cell-cell associations, we used a flow cytometry-based assay previously developed for studying contact-dependent inhibition in *E. coli* (*Ruhe et al., 2015*). In brief, stationary phase producer cells producing TdTomato and indicator cells producing Venus were mixed at a 1:1 ratio, and fluorescence measured by flow cytometry (*Figure 4C*). There were three types of events detected by the flow cytometer: TdTomato$^+$/Venus$^-$, TdTomato$^-$/Venus$^+$, and TdTomato$^+$/Venus$^+$. The latter, detected at a frequency of ~12% in the mixed population, likely represent cases in which at least one producer and at least one indicator cell are closely associated during passage through the flow cytometer. The frequency of such associated cells was not significantly different when examining mixtures of (i) two producers, (ii) an indicator and a producer, or (iii) two indicators (*Figure 4—figure supplement 1B*) suggesting that cell-cell contact is not mediated by the *cdz* genes themselves but rather by other genes that the *cdz* system effectively uses to promote killing. Cell-cell attachments were also seen at comparable frequencies in exponential phase cultures (*Figure 4—figure supplement 1C*), indicating that whatever mediates attachment is not specifically expressed in stationary phase and so likely does not contribute to stationary phase-specific killing by the Cdz system (*Figure 2A*).

## CdzC and CdzD localize to the outer membrane of producer cells

Our results suggested that CdzC and CdzD are cell associated and mediate contact-dependent killing. To further characterize the subcellular localization of CdzC and CdzD, we performed cell fractionation experiments on cells producing epitope-tagged variants of CdzC, CdzD, and CdzI (*Figures 1D* and *5A–D*). Epitope-tagged CdzC and CdzD were fully functional with respect to killing (*Figure 5—figure supplement 1A*), while epitope-tagged CdzI was partially functional as an immunity gene (*Figure 5—figure supplement 1B*). As controls for the fractionation procedure, we verified by immunoblotting that RpoA, a subunit of RNA polymerase, was found predominantly in the cytoplasmic fraction; the chemoreceptor McpA was found in the total membrane and inner membrane fractions; and RsaF$_a$, a protein involved in S-layer production, was found primarily in the total membrane and outer membrane fractions.

For HA-CdzC, there were two bands in the total membrane fraction (*Figure 5A*, S4C). One was estimated to be ~10 kDa, and likely corresponds to the unprocessed form of CdzC that includes the secretion signal peptide, while the other band likely represents the cleaved, processed form. Consistent with this interpretation, the full-length form was also seen in the inner membrane fraction, but not the outer membrane fraction, whereas the smaller form had the opposite pattern, appearing predominantly in the outer membrane fraction. Additionally, we noticed that a substantial amount of CdzC was also present in the stacking gel of the total and outer membrane fractions, suggesting that it forms larger aggregates that cannot enter the running gel. The presence of these aggregates did not depend on CdzD as they were still present in a Δ*cdzD* strain (*Figure 5—figure supplement 1D*). They were, however, dependent on the CdzAB secretion system. In a Δ*cdzAB* strain, CdzC was found in the cytosolic, total membrane, and inner membrane fractions but not the outer membrane, and no high molecular weight aggregates were seen in the total membrane fraction (*Figure 5—figure supplement 1D*). Additionally, in the Δ*cdzAB* strain, only the full-length form was detected.

We confirmed our results for HA-CdzC using a second fractionation method in which cell extracts were run through a sucrose gradient (*Figure 5B*). Immunoblots of CdzC, along with control blots for McpA and RsaF$_a$, confirmed that CdzC is found in the cell membranes, with the strongest signal seen for fractions corresponding to the outer membrane. Again, strong signal for CdzC was also

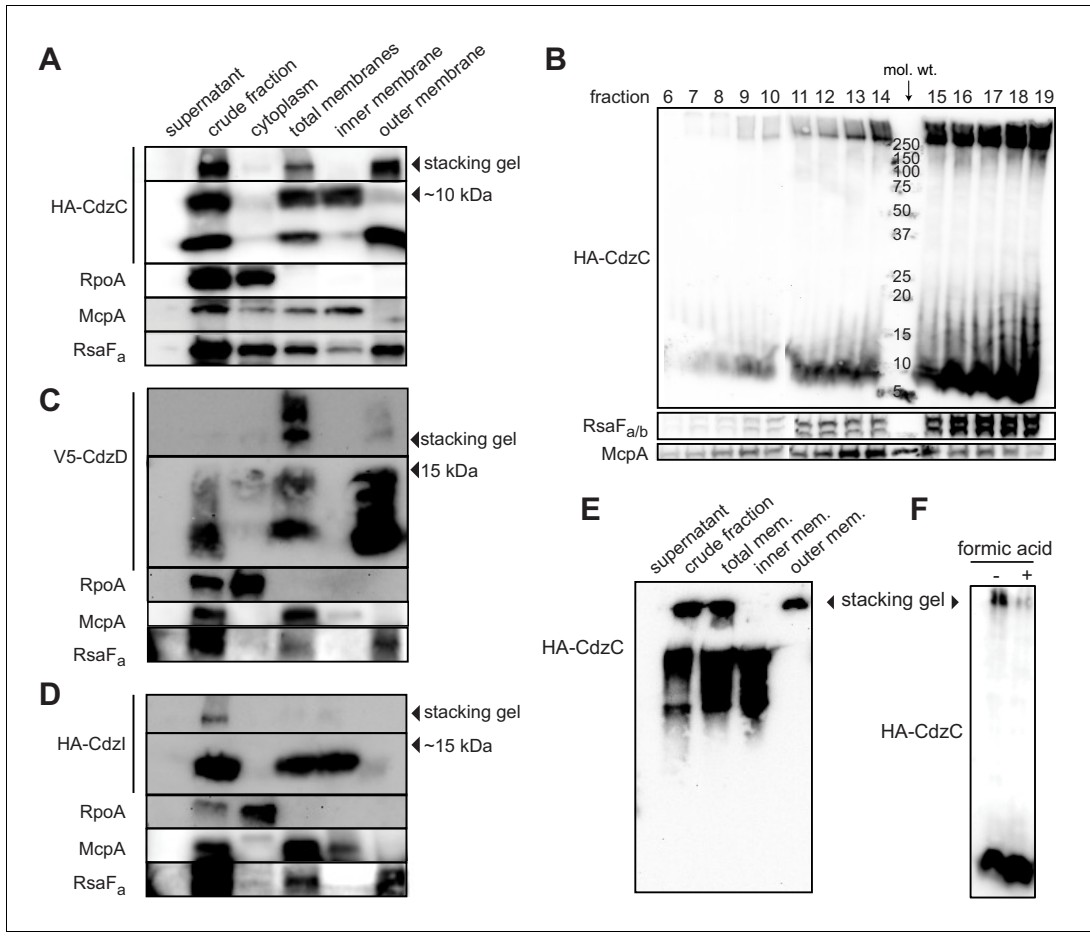

**Figure 5.** CdzC and CdzD localize to the outer membranes of cells, while CdzI is an inner membrane protein. (**A**) Western blots of the indicated cell fractions from a culture of cells producing HA-tagged CdzC and grown to stationary phase. RpoA, McpA, and RsaF$_a$ serve as controls for cytoplasmic, inner membrane, and outer membrane fractions, respectively. For HA-CdzC, the stacking gel and relevant portion of the running gel are shown. The approximate size of CdzC bands were estimated from a molecular weight ladder run in a non-adjacent lane on the same gel, and confirmed on 16.5% Tris-tricine gels (**Figure 5—figure supplement 1C**). (**B**) Western blot of the indicated fractions from a sucrose gradient (25–60% w/v) based separation of the membrane fractions from a stationary phase culture expressing HA-tagged CdzC. An approximate molecular weight ladder is shown. RsaF$_{a/b}$ and McpA serve as controls for the outer and inner membranes, respectively. (**C–D**) Same as panel A but for a strain producing (**C**) V5-CdzD or (**D**) HA-CdzI. (**E**) Western blot for HA-CdzC in Triton X-100 based cell fractionation as in panel A, but using native gels instead of SDS-PAGE. (**F**) Same as panel B, but with outer membrane fractions pooled and either treated or not with formic acid.

The following figure supplement is available for figure 5:

**Figure supplement 1.** Epitope-tagging of CdzC and CdzD does not affect their toxicity.

seen in the stacking gel indicating that CdzC forms large aggregates. Density-based fractionation methods can be misleading as insoluble cytosolic aggregates such as inclusion bodies will migrate with the heaviest fractions, often overlapping with the outer membrane fractions. We ruled out this possibility for CdzC by performing another fractionation experiment with a different range of sucrose concentrations such that outer membrane proteins would separate from the heaviest fractions (**Figure 5—figure supplement 1E**). In this experiment, CdzC was still seen in fractions corresponding to the outer membrane, as judged by the location of the control protein RsaF.

Using non-denaturing conditions and native gels, we found that nearly all of the CdzC in outer membranes remained in the stacking gel (**Figure 5E**), suggesting that CdzC mainly forms large

aggregates. These high molecular weight (>220 kDa) aggregates of CdzC were also present in denaturing gels and accounted for much of the CdzC (*Figure 5A–B*), even though those samples had been resuspended in 2% SDS and 100 mM ß-mercaptoethanol, and then boiled for 30 min. In fact, we could only solubilize these CdzC aggregates by lyophilization and the harsh addition of 90% formic acid (*Figure 5F*). Taken together, our fractionation studies indicate that CdzC predominantly forms large, insoluble aggregates in the outer membrane compartment of Cdz-producing cells.

We also examined the localization of CdzD and CdzI. Immunoblots for CdzD revealed a similar pattern as for CdzC, with both the unprocessed and mature form present in the total membrane fraction; the mature form was also abundant in the outer membrane fraction (*Figure 5C*). Like CdzC, there were also high molecular weight aggregates of CdzD in the membrane fractions. However, these aggregates were relatively faint in the outer membrane fraction, suggesting that they had been solubilized during the fractionation procedure. CdzI, the immunity protein, was found in the total membrane fraction, but in contrast to CdzC and CdzD, was primarily in the inner membrane fraction (*Figure 5D*), similar to the immunity proteins of many pore-forming bacteriocin systems (*Cascales et al., 2007*; *Lagos et al., 1999*) and those associated with some membrane-targeting contact-dependent inhibition systems secreted via type V and type VI systems (*Ruhe et al., 2013*; *Russell et al., 2014b*).

## CdzC forms large aggregates on the surface of producer cells

To assess whether CdzC forms extracellularly-exposed aggregates on the surface of intact cells, we used immunogold labeling and transmission electron microscopy on cells producing epitope-tagged CdzC. Signal from the gold-coupled antibodies was clearly associated with the surface of the cells, usually within a larger matrix of extracellular material that was likely exopolysaccharide (*Figure 6—figure supplement 1A*). Importantly, the immunogold signal was absent in exponential phase producer cells (*Figure 6—figure supplement 1B*) when CdzC is not produced at high levels and Cdz-producers cannot kill other cells.

To better visualize CdzC on the surface of cells, we also imaged producer strains lacking *manB*, a gene required for exopolysaccharide production (*Gandham et al., 2012*), and *rsaA*, which encodes the proteinaceous S-layer capsule (*Ford et al., 2007*). This strain was still competent for killing in a contact-dependent manner (*Figure 6—figure supplement 1C*). In this strain, CdzC was more clearly seen within aggregates and fibrils, ranging from 50–250 nm in diameter, that emanated from the cell surface (*Figure 6A*). Label was also seen directly on the surface of cells embedded in methylcellulose before imaging (*Figure 6—figure supplement 1D*). No gold signal was seen on the surface of cells lacking *cdzAB* (*Figure 6B*), consistent with these genes being required for secretion of CdzC. However, fibrils emanating from the cell were still present in the Δ*cdzAB* strain, suggesting that CdzC associates with these fibrils, but is not their only constituent. Taken together, our cell fractionation experiments and immunogold electron microscopy images support the conclusion that CdzC, and CdzD, localize to the surface of stationary phase cells where they mediate contact-dependent cell killing. We also confirmed localization of CdzC to the cell surface using immunofluorescence microscopy of unpermeabilized cells. Strong signal was seen for a producer strain expressing HA-CdzC, but not for an isogenic Δ*cdzAB* strain lacking the type I secretion system (*Figure 6—figure supplement 2*).

To better understand the aggregation propensity of CdzC, we examined the primary sequence of it and CdzD (*Figure 7A*). One obvious and striking feature was the abundance of glycines and small hydrophobic residues (A, V, I), which together constitute 75% of all residues in the predicted mature form of each protein. A span of 24 and 25 amino acids in CdzC and CdzD, respectively, was predicted to have high aggregation propensity (see Materials and methods). These regions of CdzC and CdzD harbor extended GxxxGxxxG motifs, or 'glycine zippers'. GxxxG motifs are found in many class IIb bacteriocins that are thought to be secreted from cells, forming transmembrane helices in target cells (*Nissen-Meyer et al., 2011*). Glycine zippers are also found in some transmembrane proteins (*Kim et al., 2005*) and they are also thought to drive the oligomerization of certain prions and amyloid-forming proteins, such as the Aß amyloid peptide in Alzheimer's disease (*Decock et al., 2016*; *Fonte et al., 2011*) and the human prion protein PrP (*Kim et al., 2005*). In other proteins with glycine zipper motifs, the x in GxxxG can be any residue, with an enrichment for hydrophobic residues (*Senes et al., 2000*). In the case of CdzC and CdzD they were exclusively small hydrophobic residues. These sequence characteristics are consistent with the aggregation propensity noted for

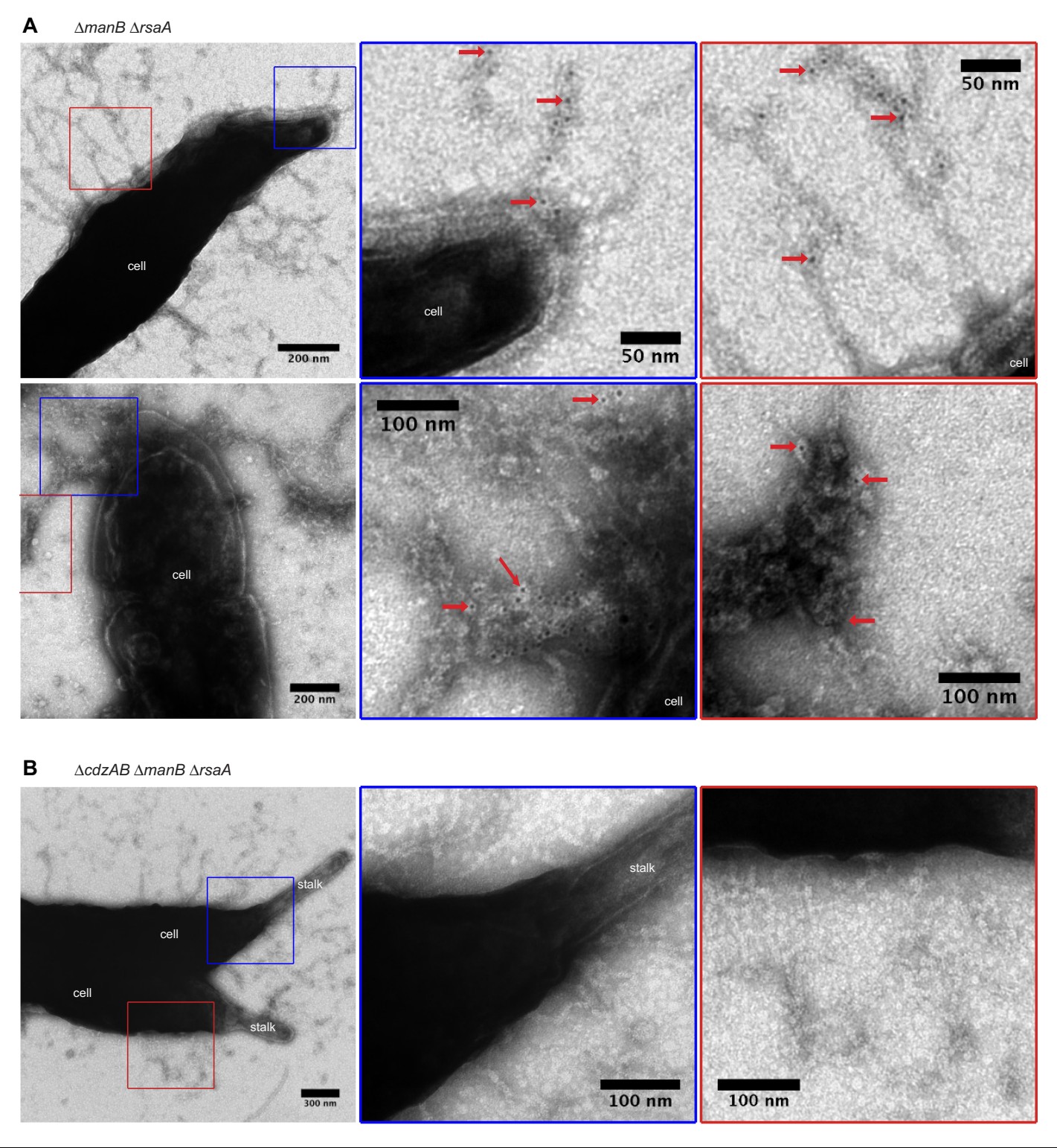

**Figure 6.** CdzC aggregates on the surface of producer cells. Immunogold labelling and transmission electron microscopy of the stationary phase producer cells indicated, each expressing an epitope-tagged CdzC. Middle and right images on each row are zoomed in regions noted on the lower magnification images on the left. Red arrows point at examples of gold particle label. Stationary phase cells of (**A**) a Δ*manB* Δ*rsaA* producer strain and (**B**) a Δ*cdzAB* Δ*manB* Δ*rsaA* strain.

*Figure 6 continued on next page*

*Figure 6 continued*

The following figure supplements are available for figure 6:

**Figure supplement 1.** CdzC aggregates on the surface of wild-type producer cells.

**Figure supplement 2.** Immunofluorescence of the producer cells indicated, each expressing an epitope-tagged CdzC.

CdzC, and also suggest that CdzC, and perhaps CdzD, insert into, and disrupt the integrity of, the membranes of target cells (*Figure 3*).

## Cdz can drive interspecies killing and Cdz-like systems are widespread in bacteria

To determine if Cdz can inhibit other species, we carried out competitions of *C. crescentus* producers against three members of the *Caulobacteraceae* family: *Caulobacter segnis*, *Brevundimonas subvibrioides* sp. Poindexter, and *Asticcacaulis exentricus* (*Abraham et al., 2014*). We also tested the ability of *C. crescentus* to inhibit three more distantly related members of the α-proteobacteria and *Escherichia coli* (*Figure 7—figure supplement 1*), none of which encode a homolog of the CdzI immunity protein. *C. segnis* was killed by a wild-type *C. crescentus* producer, but not by a *Δcdz* strain. The inhibitory effect, however, was ~3 orders of magnitude weaker than that observed with a *ΔcdzABCDI C. crescentus* indicator competed against the same producer. *B. subvibrioides* was also killed by a wild-type *C. crescentus* producer, but not by a *Δcdz* strain. In contrast, *A. excentricus* was not inhibited by *C. crescentus*, and neither were the α-proteobacteria *Rhodobacter sphaeroides*, *Sinorhizobium meliloti* and A*grobacterium tumefaciens*. Similarly, the γ-proteobacterium *E. coli* was not killed by *C. crescentus*. The Cdz-resistant species that do not encode an obvious CdzI ortholog may avoid killing because they lack a receptor or membrane composition required for the CdzC/D bacteriocin to act, or because they somehow avoid the cell-cell contact required for CdzC/D to kill.

We hypothesized the Cdz-like contact-dependent killing system is not unique to *Caulobacter*. However, BLAST searches using CdzC and CdzD as query sequences did not identify any hits, so we screened for potential Cdz-like systems in bacterial genomes using an algorithm based on gene cluster architecture and hidden Markov models (HMM) of the toxin sequences. In brief, we constructed a database of chromosomal regions that included type I secretion systems homologous to CdzAB. We then evaluated all possible small proteins within ~10 kb of the secretion system for similarity to CdzC and CdzD, using an HMM profile built from a multiple sequence alignment of CdzC, CdzD, and two Cdz-like proteins identified manually in *Bacillus sp.* YP1 and *Xylella fastidiosa* 9a5c (*Figure 7A*). The best three hits for CdzC/D-like proteins were added to the sequence alignment, which was then used as the seed to conduct an iterative search of UniProtKB using *jackhmmer* (*Eddy, 2011*). Two iterations produced a multiple sequence alignment of 239 unique hits, each exhibiting several of the key features of CdzC and CdzD: (i) 60–151 residues long, (ii) a predicted secretion leader peptide ending in G[G/A/S], (iii) one or two glycine-zipper motifs flanked, but not interrupted, by a proline or charged residue, and (iv) a short, charged or polar C-terminal region. Further iterations of this analysis began to identify proteins annotated as type IIb bacteriocins, so their results were not taken into consideration.

The phylogenetic distribution of the candidate Cdz-like systems identified (*Figure 7B*, *Supplementary file 1*) suggested that this type of contact-dependent inhibition system is widespread in bacteria, particularly in γ-proteobacteria. Notably, we identified several potential Cdz-like systems in clinical isolates of human pathogens such as *Klebsiella pneumoniae*, *Pseudomonas aeruginosa*, *Serratia marcescens*, and *Burkholderia cepacia*. Some plant pathogens of important agricultural crops also carry Cdz-like clusters, including *Xylella fastidiosa* and *Erwinia sp.*

Unlike in *Caulobacter crescentus*, most of these putative contact-dependent systems were predicted to encode a single Cdz-like toxin protein, with 14% having two adjacent proteins as in *Caulobacter*, and 5% encoding three potential toxins. The identification of immunity proteins was more difficult than the CdzC and CdzD-like proteins, although small, predicted transmembrane proteins were sometimes encoded near the Cdz-like systems identified bioinformatically. Many of the Cdz-like systems identified were likely to have been horizontally acquired as they were often flanked by

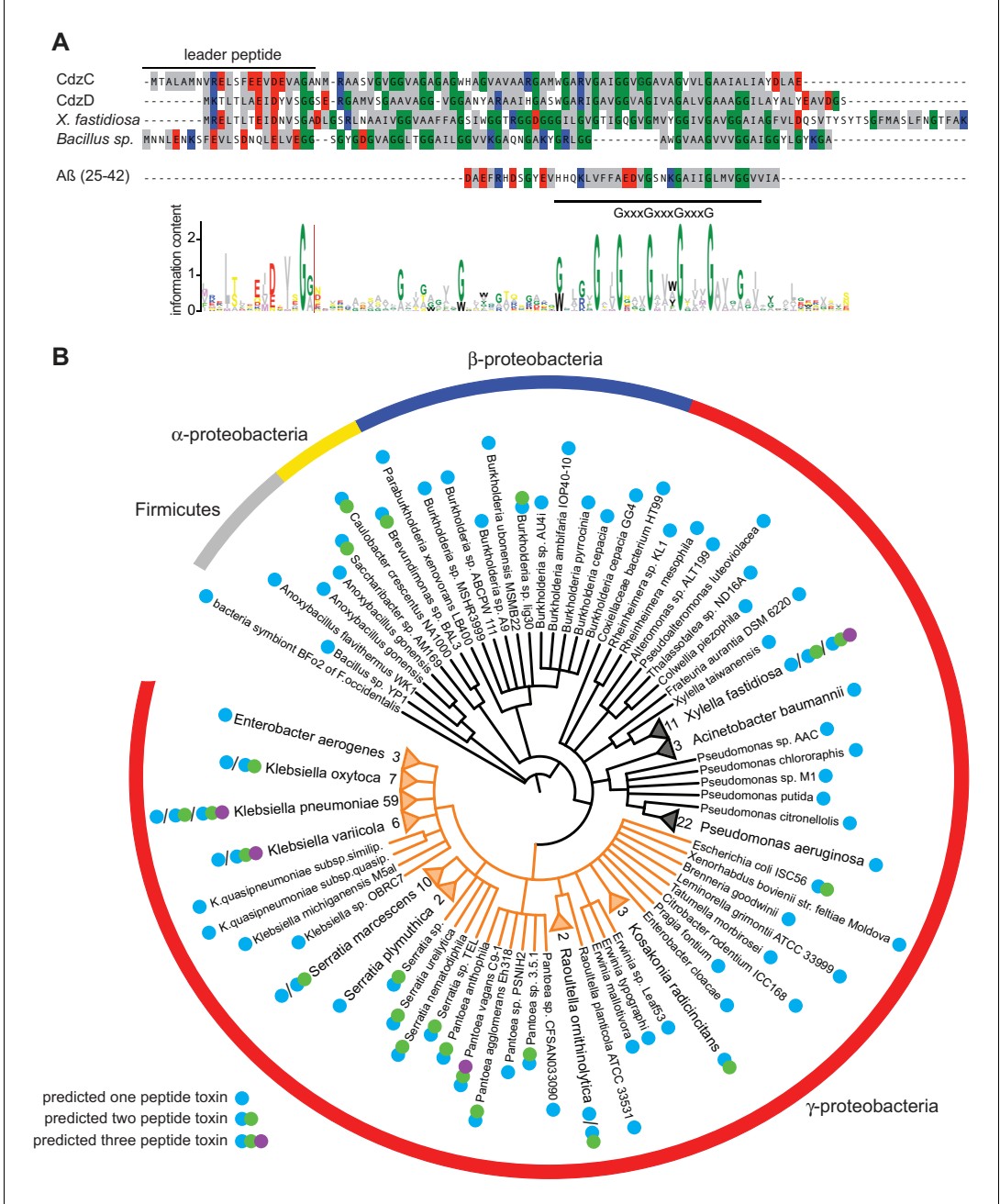

**Figure 7.** Phylogenetic distribution of putative Cdz-like systems in bacteria. (**A**) Multiple sequence alignment used as the seed in an iterative computational search for Cdz-like systems (see Materials and methods). The presumed leader peptide of each sequence is indicated. Amino acid composition is represented with colors as follows: grey, hydrophobic; glycine, green; positively charged, red; negatively charged, blue. The location of the conserved glycine zipper region is indicated below the alignment. The Aβ peptide, which also contains a glycine zipper, is shown for comparison. The sequence logo generated from all 239 Cdz-like proteins identified is shown at the bottom. (**B**) Phylogenetic distribution of Cdz-like systems identified in UniProtKB. Each branch corresponds to a unique sequence. Numbers at the tips of collapsed branches indicate the number of strains for a given species harboring a Cdz-like system. Orange branches correspond to the Enterobacteriaceae. Predicted number of Cdz-like proteins is indicated based on the legend at the bottom left.

The following figure supplement is available for figure 7:

**Figure supplement 1.** Competitions between a wild-type or Δ*cdzABCDI Caulobacter crescentus* producer and the indicator species shown: Δ*cdz Caulobacter crescentus, Caulobacter segnis, Brevundimonas subvibrioides, Asticcacaulis excentricus, Rhodobacter sphaeroides, Sinorhizobium meliloti, Agrobacterium tumefaciens,* or *E. coli.*

transposase genes, and most were present in only some isolates of a given species. Interestingly, many of the genomic regions identified, though not the *cdz* cluster in *C. crescentus*, also include genes involved in cell-cell adhesion such as thin aggregative fimbriae (Tafi), which could facilitate contact-dependent delivery of the toxins.

## Discussion

### A novel contact-dependent killing system

Our results suggest that the Cdz system in *C. crescentus* represents a new type of two-peptide bacteriocin that differs substantially from the previously described, canonical systems involving secreted, diffusible toxins. The small proteins CdzC and CdzD, which are both required for killing (*Figure 2*), were found almost exclusively in cell pellets, not cell-free supernatants. Our cell fractionation experiments indicated that CdzC and CdzD were located primarily in the outer membrane of cells, with CdzC forming large aggregates visible by electron microscopy on the surface of cells. Consistent with CdzC and CdzD remaining cell-associated, we found that these proteins only promoted the killing of other cells through direct physical contact; in a split-well set-up where indicator and producer cells shared the same growth medium but could not directly interact, no killing was observed (*Figure 4B*).

There have been other reports of possible, cell-associated bacteriocins (*Banerjee et al., 2013*; *Barbour and Philip, 2014*; *Burmølle et al., 2006*; *Daba et al., 1994*; *Dang and Lovell, 2016*; *Tahara and Kanatani, 1997*; *Yang et al., 1992*), but these prior cases are unlikely to represent contact-dependent killing systems. In most of these cases, significant killing activity was also found in cell-free supernatants and the bacteriocins were typically easily detected in supernatants, suggesting they are primarily secreted into the environment and not exclusively cell-associated as with CdzC and CdzD. Additionally, to the best of our knowledge, there has not been a direct demonstration that these previously described bacteriocins localize to the outer membranes of producing cells, or a test of whether their activity requires, or is enhanced by, cell-cell contact.

The Cdz system is very strongly upregulated upon entry to stationary phase and appears to be active only during this stage of growth as an indicator strain could be maintained at a 1:1 ratio with a producer strain throughout exponential phase, but was rapidly killed in stationary phase (*Figure 2A*). These observations suggest that *Caulobacter* cells may produce and use the Cdz system to kill neighboring cells as a means of scavenging for nutrients when they become scarce. The Cdz system was capable of killing other *Caulobacter* cells lacking the CdzI immunity protein and at least two other members of the *Caulobacteraceae* family, but not more distantly related α-proteobacteria or the γ-proteobacterium *E. coli*. A narrow spectrum of action is common for bacteriocins and may enable targeting of the most similar competitors within a given niche (*Riley, 2011*). Producing a toxin that remains cell-associated may be critical to an aquatic, oligotrophic bacterium like *Caulobacter* as a fully secreted, diffusible toxin would be quickly flushed away, providing no benefit to a producer cell. We speculate that CdzC/D, and possibly all contact-dependent inhibition systems, represent a means of ensuring benefit to producer cells, preventing the futile release of potentially costly toxins. Contact-dependent systems may also help avoid exploitation by 'cheater' cells in a population that do not pay the cost of producing a toxin while benefitting from a secreted, public good (*McNally et al., 2017*).

### Mechanism of secretion and toxin delivery

To the best of our knowledge, the Cdz system is the first contact-dependent inhibition system described involving a type I secretion system, as most rely on type V-based systems also known as two-partner secretion systems, or on type VI secretion systems (*Russell et al., 2014b*; *Willett et al., 2015*). The type V-based systems are found in the contact-dependent inhibition system first identified in uropathogenic *E. coli* and subsequently found quite broadly in other Gram-negative bacteria (*Anderson et al., 2012*; *Aoki et al., 2005*, *2010*). In that system, the CdiA and CdiB proteins use the Sec complex for translocation across the inner membrane. CdiB is then inserted into the outer membrane where it promotes the assembly of CdiA proteins on the cell surface, with subsequent delivery of the C-terminal toxin domain of CdiA to target cells (*Willett et al., 2015*). CdiA also seems to directly promote the cell-cell adhesion needed for contact-dependent inhibition. In the

case of Cdz, cell-cell attachment is not mediated by genes encoded within the *cdz* gene cluster, as both WT and Δ*cdzABCDI* strains showed similar frequencies of cell-cell adhesion by flow cytometry (*Figure 4*, S3). Thus, Cdz toxin delivery likely takes advantage of an adhesion system encoded elsewhere in the genome. Transmission electron microscopy with immunogold labeling indicated that CdzC associates with fibrils on the surface of stationary phase cells. What genes encode these fibrils and whether they mediate attachment remains to be determined. Also of note is that type I systems usually partner with an outer membrane TolC-like channel to mediate the secretion of proteins into the extracellular milieu (*Kanonenberg et al., 2013*). How the CdzA/B system manages to retain CdzC/D in the outer membrane following secretion is not yet clear and will require future studies.

## Mechanism of killing by CdzC and CdzD

Our fluorescence microscopy studies (*Figure 3*) suggest that CdzC/D kill cells lacking the immunity protein CdzI by forming pores in the membrane of target cells, a common mechanism of action for bacteriocins (*Rebuffat, 2011*). When killed, indicator cells became permeable to propidium iodide, a strong indicator that inner membrane integrity had been compromised. These cells eventually lost their characteristic rod-shape, suggesting a complete loss in cell envelope integrity. Consistent with CdzC/D being a membrane pore-forming toxin, the immunity protein CdzI was found primarily in the inner membrane, where it may directly antagonize CdzC/D function. Whether CdzC and CdzD are transferred to a target cell or whether they kill while remaining associated with the producer cell is not yet clear.

CdzC forms large aggregates that appear critical to its toxic activity. These aggregates are stable against SDS and heat, dissolvable only through harsh formic acid treatment. This propensity to aggregate likely stems from the extensive hydrophobicity of CdzC, including an extended glycine zipper region with almost entirely hydrophobic residues within each repeat (*Figure 7A*). Glycine zippers are often found in amyloid proteins (*Chang et al., 2010*; *Kim et al., 2005*) that form insoluble aggregates like those seen for CdzC. The GxxxG motif in CdzC and CdzD have also been seen in bacteriocins thought to kill target cells by inserting into their membranes (*Kim et al., 2005*). Thus, the extended glycine zipper region of CdzC may mediate both its aggregation on the surface of producer cells and its insertion into the membrane of target cells. In this vein, we note that some annotated type IIb bacteriocins harbor GxxxG motifs (*Nissen-Meyer et al., 2011*) and may, like CdzC, form large aggregates on the surface of producing cells to drive contact-dependent killing; however, as noted earlier, no such analyses have been reported for bacteriocins. The Cdz system may represent a special sub-class of type IIb bacteriocins or a new class.

CdzD has a similar set of sequence characteristics as CdzC, including the glycine zipper repeats with almost entirely hydrophobic residues. However, CdzD aggregates appear to be less stable as they were easily solubilized in our cell fractionation studies. However, CdzD and CdzC were both fully required for killing (*Figure 2*). CdzD could help promote CdzC secretion or it may seed CdzC aggregates, although CdzC still formed very high molecular weight aggregates in a Δ*cdzD* strain (*Figure 5—figure supplement 1D*). Alternatively, CdzD may participate only in the 'warhead', or pore-forming element, of a CdzC-driven aggregate, creating a pore alone or in combination with CdzC. Notably, CdzC and CdzD need to be secreted to be active as a strain lacking the secretion system, CdzAB, was fully viable. It is possible that CdzC and/or CdzD must be modified upon secretion. Although the *cdz* gene cluster does not encode any obvious modification enzymes, as is often the case for bacteriocin gene clusters (*James et al., 1992*; *Nissen-Meyer et al., 2011*), CdzC and CdzD may at least need to have their leader peptides removed for full activity as toxins or possibly to aggregate, with only aggregates being active.

CdzC was associated with fibrils on the surface of fixed cells, although CdzC is likely not the only, or even primary, component of these fibrils. As already noted, these fibrils may mediate cell adhesion, with the Cdz system effectively using them for toxin delivery. Associations between bacteriocins and fibrils have been reported previously. The pore-forming microcin MccE492 from *Klebsiella pneumoniae* RYC492 is thought to form amyloid-like, but non-toxic fibrils on the surface of cells in stationary phase (*Bieler et al., 2005*). The toxic form of MccE492 is non-aggregated and diffusible, such that its inhibitory activity is seen only in the supernatants of exponential phase cells. In *Staphylococcus aureus*, phenol soluble modulins (PSM), which share some similarity to eukaryotic antimicrobial peptides, form amyloid-like fibrils required for biofilm stabilization. When purified, some PSM variants have antibacterial activity (*Cogen et al., 2010*; *Schwartz et al., 2012*).

An association between small antimicrobial proteins and amyloid formation has also been described in the case of several eukaryotic peptides. En route to forming mature fibers, many amyloid proteins pass through a toxic oligomeric stage that forms pores in membranes (*DePas and Chapman, 2012*). The most notable examples are the human major prion protein PrP and the amyloid-beta (Aß) peptide that accumulates in Alzheimer's disease, which has a glycine zipper region similar to those in CdzC and CdzD (*Figure 7A*). The small hydrophobic residues in the zipper region of the Aß peptide are thought to promote ß-aggregation by creating a 'steric-zipper' (*Chang et al., 2010*). A leucine substitution in one of the glycine residues is sufficient to significantly alter cytotoxicity and oligomer formation by the Aß peptide (*Decock et al., 2016*; *Fonte et al., 2011*).

## Cdz-like bacteriocins may be widespread in proteobacteria

Collectively, our studies point to the existence of a new family of contact-dependent bacteriocins, secreted via a type I secretion system. This family appears to be relatively widespread, particularly in proteobacteria, with each system containing one to three small (~5–10 kDa) glycine zipper-based toxins. Many of these systems appear to have been horizontally acquired as even closely related species sometimes lack an orthologous copy. Additionally, many of these putative Cdz-like systems are part of, or adjacent to, genomic islands that include genes often associated with recent lateral transfers, such as transposases, CaaX proteases, and fimbrial adhesins.

In some cases, closely related species or strains do have orthologous Cdz-like systems, although the similarity is usually highest for the secretion systems, with substantial variation between the Cdz-like proteins. This type of inter-strain variation has been observed in other types of contact-dependent systems where they may impact self/non-self recognition, thereby playing a critical role in shaping community structure (*Alcoforado Diniz and Coulthurst, 2015*; *Anderson et al., 2012*; *Russell et al., 2014a*, *2014b*).

The Cdz-like systems we identified bioinformatically are not restricted to organisms from a specific ecological niche and instead appear to be present in a range of bacteria with diverse lifestyles. Of particular interest are the Cdz-like systems present in clinical isolates of human pathogens. The importance of outcompeting commensal bacteria to establish or maintain an infection has been postulated for several pathogens (*Hecht et al., 2016*; *Russell et al., 2014b*) and the Cdz-like systems identified in pathogens (*Figure 7*) could play such a role. Characterizing these systems and showing whether they indeed mediate contact-dependent inhibition is an important future goal.

## Final perspective

*Caulobacter* is a well established model for understanding cell-cycle regulation in bacteria. Considerably less attention has been placed on understanding how it thrives in nutrient-poor aquatic conditions, and no social behaviors or quorum-sensing systems have been described to date. Here, we identified an atypical, but potentially widespread bacteriocin-like system that likely enables *Caulobacter* to compete with other strains and some other species, by killing nearby cells lacking the Cdz system. The Cdz contact-dependent inhibition system presumably also allows *Caulobacter* to avoid the dilemma of producing an expensive common good that could get easily washed away in aquatic environments or that would render a population susceptible to cheaters. Finally, the discovery of the Cdz system, which uses a type I secretion system, expands the repertoire of contact-dependent inhibition mechanisms and suggests that such systems may be even more widespread than previously appreciated.

# Materials and methods

## Bacterial strains and growth conditions

All oligonucleotides, strains and plasmids used in this study are listed in *Supplementary files 2*, *3*, and *4*, respectively, along with details of how each strain was constructed. *E. coli* strains were grown at 37°C in LB medium. *A. excentricus* was grown in PYE at 24°C. *A. tumefaciens*, *S. meliloti*, and *R. sphaeroides* were grown in LB at 30°C for propagation and plating experiments, and in PYE for competitions with *C. crescentus*. *C. crescentus*, *C. segnis* and *B. subvibrioides* strains were grown at 30°C in PYE. Except when specified otherwise, stationary phase cells and culture supernatants were harvested 4 hr after reaching an OD$_{600}$ of 0.85. Media were supplemented, as necessary, with

a final concentration of 500 μM vanillate (pH 7.5 stock in water), 1 μM IPTG, or 100 μM cumate (4-isopropylbenzoic acid, stock in 100% ethanol) to induce expression from the P$van$, P$lac$, and P$_{PQ5}$ promoters, respectively. Antibiotics were used at the following concentrations (liquid/solid media for *E. coli*; liquid/solid media for *C. crescentus*; in μg/ml): oxytretracycline (12/12; 1/2), kanamycin (30/50; 5/25), gentamycin (15/20; NA/5). For *C. segnis* and *A. tumefaciens*, plates were supplemented with 50 μg/ml kanamycin. For *S. meliloti*, plates were supplemented with 200 μg/ml neomycin.

## Competition experiments

For time courses of competitions between different strains (*Figure 2A*), exponential phase cultures of each strain were diluted to OD$_{600}$ = 0.025 and incubated at 30°C for 30 min (min) before mixing at a 1:1 ratio by volume. Every hour aliquots were taken and 10-fold serial dilutions (10 μL each) spotted on PYE plates containing antibiotics appropriate for selection of each competitor; colony forming units (CFU) were counted after two days. For end-point competitions (*Figure 2C–F*), each competitor was grown separately to an OD$_{600}$ = 0.85, at which point they were mixed at a 1:1 ratio by volume. For strains harboring P$_{van}$ constructs, vanillate was added 5 hr prior to mixing strains. CFUs for each competitor were quantified as described above at the time of mixing, and then 4 and 16 hr in competition. The competitive index was calculated by taking the ratio of the two strains at the final time point, normalized by the ratio of the two strains at t = 0 (see *Figure 2B*). To test the effect of growth phase on killing, each competitor strain was grown separately until exponential phase (OD$_{600}$ = 0.2) or stationary phase (OD$_{600}$ = 1). For competitions, 10 mL of the exponential phase culture were pelleted (6000 x g, 2 min) and cells were resuspended in 1 mL of the stationary phase culture.

## Testing for inhibitory activity in culture supernatants

To test the effects of cell-free supernatant on an indicator strain, the supernatant from a stationary phase culture of a producer strain was recovered by pelleting cells twice in succession at 13,000 x g for 2 min and filtering through a low protein binding 0.45 μm HT Tuffryn membrane (Pall Life Sciences). This filtered supernatant was used to resuspend stationary phase indicator cells pelleted by centrifugation, and their survival was quantified after incubation at 30°C for 4 or 16 hr. As a control, supernatant from an indicator strain was tested, allowing us to quantify background rates of cell death in stationary phase under the experimental conditions used.

To concentrate supernatants from stationary phase cultures of producer strains, we used trichloroacetic acid (TCA) precipitation of total protein in culture supernatants using an adaptation of a published protocol (*Koontz, 2014*). In brief, 2 L of culture supernatant was centrifuged twice at 13,000 x g for 10 min to remove cells. Samples were then resuspended to a final concentration of 10% (w/v) TCA and incubated at room temperature for 20 min followed by incubation at −20°C overnight. Samples were then centrifuged at 15,000 x g for 20 min. Pellets were washed three times with 7 mL of −20°C acetone, vortexing to resuspend and incubating 2 hr (first wash) or 30 min (second and third wash) at −20°C, and then centrifuging at 15,000 x g and incubating at 4°C for 10 min. The final pellet was air-dried and resuspended in PYE, and tested by immunoblotting. To test the activity of the TCA-precipitated proteins in liquid culture, the bulk protein pellet was resuspended in 2 mL of supernatant from a stationary phase indicator culture in PYE and mixed with indicator cells to test their survival.

Bulk precipitation of proteins in culture supernatants using ammonium sulfate was done following a procedure outlined previously (*Simpson, 2006*). The starting sample was 50 mL of stationary phase culture supernatant, which was filtered using a 0.22 μm Steriflip PES vacuum filter (EMD Millipore). After ammonium sulfate addition up to 85% saturation at 0°C and 1 hr equilibration, samples were pelleted at 17,000 x g for 1 hr. Pellets were then resuspended in 500 μL of 10 mM Tris-HCl pH 7.4 and desalted using an Illustra NAP-5 column (GE Healthcare) following the manufacturer's protocol. Eluate was concentrated to 150 μL using an Amicon Ultra-4 3 K centrifugal filter (EMD Millipore) spun at 7500 x g for 10 min.

To rule out the possibility that large diffusible aggregates present in the supernatant were pelleting with the cells during centrifugation steps, we performed competitions in which the stationary phase producer cells were killed by heating for 5 min at 98°C, or 4 hr of treatment with 4 μg/ml gentamicin, prior to mixing without centrifugation to an equal volume of stationary phase indicator

carrying a gentamicin resistance cassette. Indicator survival was quantified after incubation for 16 hr at 30°C.

## Split-well competition experiments

Sterile single-well Millicell hanging inserts with a PET membrane (Millipore, 0.4 or 8 μm pore size) were placed into 6-well polystyrene plates to split the well into upper and lower chambers. Only three wells per plate were used to prevent cross-contamination between adjacent wells. Each competitor strain was grown separately to $OD_{600}$ = 0.85 and 3 mL added to the top or bottom of the split-well. Plates were covered with a lid and incubated at 30°C with shaking (90 rpm to prevent splashing between top and bottom of the well) and the competition evaluated as above. For each pair of strains, independent competitions were done with strains swapped between upper and lower chambers, to rule out effects due to position within the well.

## RNA-seq and transcriptional reporters

Exponential phase RNA-seq data (PYE) were taken from GEO sample GSM1326109 (*Schrader et al., 2014*). For stationary phase, RNA-seq was performed as previously described, with a few modifications (*Guo et al., 2014*). *Caulobacter* cells were grown in PYE to $OD_{600}$ = 1.4 and 2 mL of cells were harvested in the presence of 1 mL 95% ethanol/5% phenol and frozen in liquid nitrogen. The cell pellet was extracted by hot phenol method (*Masse et al., 2003*) and ribosomal RNA removed with the MICROBExpress kit (Life Technologies). tRNAs were not removed to ensure recovery of small RNAs. Samples were converted to a sequencing library by lightly fragmenting the RNA with RNA fragmentation reagents (Ambion), isolating and dephosphorylating 25–50 mers with T4 PNK (New England Biolabs), ligating linker-1 (Integrated DNA Technologies) to fragments with T4 RNA ligase II (Epicentre). After reverse transcription was performed with Superscript III (Invitrogen) and RT_oligo, the RNA was hydrolyzed with NaOH, and the ssDNA cDNA was circularized with CircLigase (Epicentre). Final sequencing libraries were PCR amplified with Phusion polymerase and adapter oligos (o231 and o230). Single-end sequencing was performed on Illumina HiSeq2000 (BioMicroCenter MIT) and aligned to NC011916.1 (NA1000) with bowtie (*Langmead et al., 2009*) allowing for one mismatch. Mapped reads were normalized by total number of reads (RPM) and displayed with IGV_2.3.40 software using the 'mean' windowing function. Gene expression data are available in the Gene Expression Omnibus: GSE96582.

For time-courses of the *cdzC* transcriptional reporter, an overnight culture was diluted to $OD_{600}$ = 0.025 in PYE. For three replicate cultures, cells were harvested every hour and spotted into 1.5% agarose pads for imaging and fluorescence quantification with MicrobeJ (*Ducret et al., 2016*). Fold induction was calculated relative to the fluorescence density at $OD_{600}$ = 0.025.

## Immunoblotting

Cells were pelleted by centrifugation and resuspended in 2X SDS sample buffer, heated to 95°C for 30 min and equivalent $OD_{600}$ amounts separated on Mini-PROTEAN TGX 15-well Any kD$^{TM}$ Tris-HCl gels (Bio-Rad) at 150 V using the Laemmli buffer system (*Laemmli, 1970*). For 16.5% tris-trisine gels (Mini-PROTEAN, Bio-Rad), glycine was replaced in the running buffer with 0.1M tricine. Transfer was done at 100 V for 100 min using a 0.2 μm Immuno-Blot PVDF membrane (Bio-Rad), Towbin buffer and the Mini Trans-Blot Module (Bio-Rad). The following antibodies were used at a 1:5000 dilution: anti-RpoA (Neoclone #W0003, mouse), anti-McpA ((*Iniesta et al., 2006*), rabbit), anti-RsaF ((*Ford et al., 2007*), rabbit), anti-HA (Cell Signaling Technology C29F4, rabbit), anti-V5 (Cell Signaling Technology D3H8Q, rabbit), HRP-conjugated goat anti-rabbit (ThermoFisher), and HRP-conjugated goat anti-mouse (ThermoFisher). Membranes were exposed to SuperSignal West Femto Maximum Sensitivity Substrate (Thermo Fisher) and imaged with a FluorChem M Imager (ProteinSimple).

For non-denaturing gels, the following modifications were done: samples were resuspended in 1X sample buffer (62.5 mM Tris-HCl, pH 6.8, 20% glycerol, 0.005% bromophenol blue) and directly loaded without heating; running buffer was 25 mM Tris, 192 mM glycine. Transfer was done in two steps, using running buffer: 1 hr at room temperature (with ice block) with constant current 400 mA, followed by replacement of running buffer with ice-cold fresh buffer and transfer for two more hours at a constant voltage of 100 V at 4°C.

## Fluorescence microscopy of competitions

For time-course imaging of competition experiments, cultures were set up as described above, with the addition of IPTG to the starting overnight culture and to the competition flasks to induce fluorescent reporter expression. At each time point, cells were harvested, diluted in one volume PYE and propidium iodide added to a final concentration of 15 µM. Cells were incubated in the dark for 15 min, spotted onto PYE 1.5% agarose pads and imaged immediately. Phase contrast and fluorescent images were taken on a Zeiss Axiovert 200M microscope with a 63× phase or αFluar 100×/1.45 oil immersion objective, using a digital camera (Orca-II ER; Hamamatsu Photonics) and Metamorph software (Universal Imaging, PA). The following emission/excitation filters were used: YFP, 500/25 and 535/30m; CFP, 436/25 and 480/40; mCherry, 560/40 and 630/75. Image analysis to quantify fluorescent signals for each cell was calculated using MicrobeJ (*Ducret et al., 2016*). Three independent replicates were analyzed, with a minimum of 250 cells per frame. Fold induction for the $P_{cdzC}$-YFP transcriptional reporter was calculated relative to fluorescence density at $OD_{600}$ = 0.025. Image overlays were generated using Fiji (*Schindelin et al., 2012*).

## Immunogold labeling and transmission electron microscopy

Cells from a PYE culture were treated with 0.1% paraformaldehyde for 15 min and allowed to sediment for 15 min into formvar/carbon-coated nickel electron microscopy grids freshly treated with 0.2% paraformaldehyde. After 15 min blocking in 1% (w/v) BSA in PBS, grids were moved to anti-HA antibody (1:50 dilution in blocking buffer) for 2 hr, followed by 1 hr in 6 nm Colloidal Gold secondary antibody (Donkey Anti-Rabbit IgG from Jackson ImmunoResearch, 1:30 dilution in blocking buffer), and then three 1 min washes in PBS. After a brief water wash, negative staining was performed using 1% uranyl acetate. Grids were blot-dried and imaged on a FEI Technai Spirit Transmission Electron Microscope at 80 kV. For methylcellulose embedded samples, after the final water wash, grids were incubated for 10 min in 1% neutral uranyl acetate, briefly washed in water and floated on a 0.2% methylcellulose/3%Polyvinyl alcohol solution containing 1% uranyl acetate. Excess embedding solution was drawn off and grids were air-dried.

## Immunofluorescence

Cells were labeled with anti-HA antibodies as described previously (*Nomellini et al., 2010*), with the following modifications: 1 hr fixation with 2.5% paraformaldehyde at room temperature, 1 hr incubation on ice in primary antibody (2 µL antibody to 200 µL final volume, anti-HA Cell Signaling Technology C29F4), 40 min on ice in secondary antibody (1 µL antibody to 100 µL final volume, Life Technologies Alexa Fluor 555 goat anti-rabbit IgG), and two additional wash steps after primary and secondary antibody incubations (done by centrifugation and resuspension in 1.5 ml PYE). After the final wash, cells were resuspended in 100 µL Slowfade Diamond Antifade (Life Technologies) and spotted onto 1.5% agarose pads for imaging.

## Flow cytometry

Cultures were grown to $OD_{600}$ = 0.6 (stationary phase) or $OD_{600}$ = 0.025 (exponential phase), supplemented with cumate to induce the expression of Venus or Tdtomato, and incubated for 5 hr at 30°C with shaking. Exponential phase cells were diluted to $OD_{600}$ = 0.04, and the cells expressing Venus or tdTomato were mixed at a 1:1 ratio and cultured for 2 hr or immediately analyzed. Cells were diluted 1:50 into PBS, homogenized by inversion, and flow cytometry performed for 50,000 cells per strain mixture, with an event rate of ~12,000 events/µL. The following settings were used on an BD LSR-II cytometer: FITC voltage 500; PE YG voltage 500; SSC 300; FSC 324. Venus and tdTomato were excited using 488 nm or 561 nm lasers and detected using 515/20 nm or 582/42 nm filters, respectively. Data were visualized using FlowJo software.

## Subcellular fractionation

The spheroplasting and Triton X-100 cell fractionation protocol was adapted from *Gray et al. (2011)*. The initial cell pellet was washed once in an equal volume of PYE by resuspension and centrifugation to remove supernatant protein contamination. Lysozyme was used at a final concentration of 100 µg/ml and 10 freeze-thaw cycles were carried out, alternating between liquid $N_2$ and 50°C. Lysates were treated with 5 µg/mL DNase I with 25 mM $MgCl_2$. Incubation in 2% (w/v) Triton X

solubilization buffer was done overnight at 37°C with mixing. All cell fractions were mixed 3:1 with 4X SDS or native sample buffer, and the final insoluble fraction was resuspended in 1X sample buffer.

Cell lysis and sucrose gradient separation of membrane proteins was conducted as previously described (*Anwari et al., 2010*), supplementing the lysis buffer with 25 µg/mL DNase I and 10 mM MgCl$_2$. Total membranes were resuspended in 25 or 35% (w/v) sucrose in 5 mM EDTA, and separated on either of two different six step sucrose gradients set up on Beckman ultra-clear tubes (14 × 89 mm): 35:40:45:50:55:60% or 45:50:55:60:65:70% (w/v) sucrose in 5 mM EDTA. 500 µL fractions were collected using a BioComp Piston Gradient Fractionator and mixed with 4X SDS sample buffer for immunoblot analysis.

For formic acid extractions of CdzC aggregates, outer membrane fractions determined by immunoblotting to contain CdzC were pooled together, mixed with 5 volumes of TEM buffer (10 mM Tris pH 7.5, 1 mM EDTA, 10 mM MgCl$_2$,10% glycerol) and pelleted at 221,000 x g and 4°C for 1.5 hr with an ultracentrifuge containing a SW41 rotor. Pellets were resuspended in 90% formic acid, frozen in liquid N$_2$, lyophilized and resuspended 1X SDS sample buffer for immunoblot analysis.

## Computational prediction of aggregation and searches for Cdz-like systems

The protein aggregation propensity and amylogenic regions of CdzC, CdzD and CdzI were predicted based on amino acid sequence using AmylPred 2 (*Tsolis et al., 2013*), WALZ (high specificity threshold) (*Oliveberg, 2010*), MetAmyl (with high specificity threshold) (*Emily et al., 2013*) and PASTA 2.0 (with peptides threshold) (*Walsh et al., 2014*). A sequence region was considered aggregation prone if at least three of the above algorithms were in accordance.

To identify CdzC/D-like proteins in other organisms, a multiple sequence alignment was done using CLUSTALW with CdzC, CdzD, and two Cdz-like proteins identified by the GFIT function of KEGG: *Bacillus sp*. YP1 (QF06_18395) and *Xylella fastidiosa* 9a5c (XF_0262). The alignment was used to generate a Hidden Markov Model (HMM) profile using the hmmbuild function in HMMER3 (*Eddy, 2011*). Separately, a *blastp* search was done against bacterial (taxid:2) sequences in the NCBI non-redundant nucleotide (nr) database to identify homologs of CdzB. For 639 hits obtained, flanking regions 10 kb upstream and downstream were downloaded and all possible ORFs between 60 and 200 residues were translated. This ORF list was scanned with *hmmsearch* using the HMM profile built. The best three hits were added to the multiple sequence alignment. This updated alignment was used as an input for an iterative search of the UniProtKB database using *jackhmmer* (*Eddy, 2011*), with the bias composition filter off. After the first iteration, hits that included proline or cysteine residues interrupting the glycine-zippers or that had spans of polar/charged residues were filtered out before carrying out a second iteration. Results that did not have a type I secretion system in the 10 kb flanking region were also dropped from the final list. For generation of the Cdz-like protein phylogenetic distribution tree, taxonomic information of each hit was downloaded from the NCBI Taxonomy Browser, and the tree generated using iTOL (*Letunic and Bork, 2016*).

## Acknowledgements

We thank the labs of T Le, L Shapiro, J Smit, S Bell, R Sauer, A Grossman, and Y Brun for strains and reagents. We acknowledge G Paradis at the Koch Institute Flow Cytometry Core at MIT for assistance with flow cytometry and especially N Watson at the Whitehead Institute Keck Microscopy Facility for help with the electron microscopy. We thank M LeRoux for helpful discussions and comments on the manuscript. This work was supported by an NIH grant (R01GM082899) to MTL, who is also an Investigator of the Howard Hughes Medical Institute, and by an HHMI International Predoctoral Fellowship to LG.

## Additional information

### Competing interests

MTL: Reviewing editor, *eLife*. The other authors declare that no competing interests exist.

## Funding

| Funder | Grant reference number | Author |
|---|---|---|
| National Institutes of Health | R01GM082899 | Michael T Laub |
| Howard Hughes Medical Institute | | Leonor García-Bayona<br>Michael T Laub |

The funders had no role in study design, data collection and interpretation, or the decision to submit the work for publication.

## Author contributions

LG-B, Conceptualization, Data curation, Formal analysis, Validation, Investigation, Visualization, Methodology, Writing—original draft, Writing—review and editing; MSG, Data curation, Investigation, Methodology; MTL, Conceptualization, Formal analysis, Supervision, Funding acquisition, Investigation, Writing—original draft, Project administration, Writing—review and editing

## Author ORCIDs

Michael T Laub, http://orcid.org/0000-0002-8288-7607

## Additional files

### Supplementary files

• Supplementary file 1. Cdz-like peptides identified in UniprotKB.

• Supplementary file 2. Oligonucleotides.

• Supplementary file 3. Strains.

• Supplementary file 4. Plasmids.

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
