## [Decision Letter]

Thank you for submitting your article "Contact-dependent killing by *Caulobacter crescentus* via cell surface-associated, glycine zipper proteins" for consideration by *eLife*. Your article has been favorably evaluated by Gisela Storz (Senior Editor) and three reviewers, one of whom is a member of our Board of Reviewing Editors. The following individual involved in review of your submission has agreed to reveal his identity: Christopher Hayes (Reviewer #3).

The reviewers have discussed the reviews with one another and the Reviewing Editor has drafted this decision to help you prepare a revised submission.

This work introduces the first bacteriocin system that does not rely on diffusible toxins but instead on toxins that remain anchored to the cell surface of producer cells thereby facilitating the killing of neighboring cells through cell-cell contact. This new mechanism expands the range of antimicrobials and bacterial weapons used by bacteria to compete across species boundaries. While all known bacterial bacteriocins rely on killing through diffusion, this system depends on cell-cell contact, a mechanism that maybe optimized for aqueous environments and that helps to avoid fitness problems arising from the distribution of public goods in bacterial communities.

Although all reviewers are enthusiastic about this work, they consider that some aspects of the manuscript could be improved, mainly the evidence that the mechanism is contact-dependent.

i) First, the finding that cells tend to aggregate per se is not a strong argument for contact-dependent killing. Likewise, results from split-well experiments, although in full agreement with contact dependency, could also be interpreted differently (e.g. retention of large diffusible aggregates and fibril material by small pore filters). Contact-dependent killing could be shown directly by simple imaging techniques, e.g. when producers and indicators are mixed on agar pads or in microfluidic devices in a 1:10 or 1:100 ratio one would expect to observe time-dependent killing primarily in the neighborhood of producers. A simple statistical analysis of the spatial distribution of producers and killing events could help build support for a killing mechanism involving cell-cell contact.

Time-lapse microscopy could also be indicative of the killing process. For example, it is reasonable to suggest that Cdz is a pore-forming system, but this only relies on PI permeability, which could be indirect.

ii) The fibrils are intriguing: they appear not to be EPS or S-layer constituents (they are absent from an *rsaA manB* mutant). Could these structures be artefacts of the EM (i.e. if the cells are stressed)? Are they observed in EM section, by cryo-EM or by live microscopy using various stains? The authors could easily stain the Cdz proteins with the available fluorescent antibodies to show that surface exposure can be detected by a complementary approach. Last, are there any known adhesins or fimbriae candidates also upregulated in the RNA-seq dataset during stationary phase?

iii) The proline substitution mutants (Figure 7—figure supplement 1) are quickly glossed over, compared to the rest of the study. An alternative view is that the Pro mutant is unstable or fails to be exported. To exclude that and to substantiate the possibility that the zipper is important for killing activity as opposed to export/stability the authors should monitor secretion/surface exposure of this mutant variant. In absence of more controls the use of these mutants is not essential for the demonstration.

iv) In their interspecies killing assays with *Caulobacter crescentus* Cdz producers, the authors have used two closely related α-proteobacteria. Have they used more distantly related members of this taxonomic group and can they comment on the phylogenetic distribution of potential Cdz prey? The Cdz mnemonic implies that only glycine zipper-like toxins are deployed by T1SS. Did the authors uncover other recognizable toxin-immunity genes linked to cdzAB related loci? Could this secretion mechanism be used to transfer nucleases, lipases or murein hydrolases into recipient cells?

---

## [Author Response]

*[…] Although all reviewers are enthusiastic about this work, they consider that some aspects of the manuscript could be improved, mainly the evidence that the mechanism is contact-dependent.*

i) First, the finding that cells tend to aggregate per se is not a strong argument for contact-dependent killing. Likewise, results from split-well experiments, although in full agreement with contact dependency, could also be interpreted differently (e.g. retention of large diffusible aggregates and fibril material by small pore filters). Contact-dependent killing could be shown directly by simple imaging techniques, e.g. when producers and indicators are mixed on agar pads or in microfluidic devices in a 1:10 or 1:100 ratio one would expect to observe time-dependent killing primarily in the neighborhood of producers. A simple statistical analysis of the spatial distribution of producers and killing events could help build support for a killing mechanism involving cell-cell contact.

*Time-lapse microscopy could also be indicative of the killing process. For example, it is reasonable to suggest that Cdz is a pore-forming system, but this only relies on PI permeability, which could be indirect.*

We agree that cell-cell aggregation/adhesion alone is not a strong argument for contact- dependent killing. However, as noted in the manuscript, we performed these experiments as some frequency of cell-cell adhesion is presumably necessary for a contact-dependent killing mechanism in culture. With regard to the split-well experiments: we also agree that such an experiment in isolation would be only consistent with a contact-dependent mechanism, but would not rule out the possibility noted by the reviewers that large diffusible aggregates released from cells are mediating the killing and get stuck in the filters of the split well. However, importantly, it must be recalled that (i) our immunoblotting experiments (Figure 1) failed to detect CdzC and CdzD in culture supernatants, (ii) our cell fractionation experiments, EM, and new immunofluorescence data (see point ii below) demonstrate that CdzC is present on the surface of cells, and (iii) culture supernatant alone, even concentrated up to 2000X, showed no killing activity (Figure 4). With regard to this last point about supernatants, one could argue that the generation of cell-free supernatants requires a centrifugation step and that large, diffusible aggregates could have pelleted with cells. To rule out this possibility, we performed two additional experiments. First, we pre-treated gentamicin-sensitive producer cells with gentamicin for 4 h before adding them to indicator cells, without centrifugation. No killing was observed indicating that large, diffusible toxin aggregates are likely not present in the supernatant, and that producer cells must be alive to kill (see new Figure 4). Second, we boiled producer cells before adding them and their supernatant, again without a centrifugation step, to indicator cells. If large, diffusible aggregates were present in the supernatant, this preparation should still kill indicator cells, but it did not (see new Figure 4). In sum, it is our collective set of results, not the split-well experiment in isolation, that strongly favors a contact-dependent mechanism. We emphasize this point in the revised manuscript (see subsection “Cdz-based killing requires physical contact between producer and indicator cells”). We would also note that a demonstration of contact-dependent killing by time-lapse microscopy is something that still has not, to the best of our knowledge, been performed for the CDI system in *E. coli*, as such studies are difficult, and potentially even more difficult for an oxygen-sensitive organism like *Caulobacter crescentus* that cannot be monitored on agar pads for extended periods of time. This is an important challenge for the future, but goes way beyond the scope of this paper, which, as noted, already uses several different experiments to collectively support the notion of Cdz being a contact-dependent system.

*ii) The fibrils are intriguing, they appear not to be EPS or S-layer constituents (they are absent from an rsaA manB mutant). Could these structures be artefacts of the EM (i.e. if the cells are stressed)? Are they observed in EM section, by cryo-EM or by live microscopy using various stains? The authors could easily stain the Cdz proteins with the available fluorescent antibodies to show that surface exposure can be detected by a complementary approach. Last, are there any known adhesins or fimbriae candidates also upregulated in the RNA-seq dataset during stationary phase?*

The cells imaged by EM do not show any of the typical features of stressed *Caulobacter crescentus* cells, but we cannot, of course, know the exact phenotypic state of these cells immediately before imaging by EM. Nevertheless, the key issue here is whether CdzC can be visualized on the cell surface by a complementary approach. To address this issue, we have now performed immunofluorescence microscopy using our strain expressing epitope-tagged CdzC, without permeabilizing cells such that staining should only occur if CdzC is expressed on the surface of cells. As shown in the new Figure 6—figure supplement 2, CdzC-HA is indeed clearly stained by immunofluorescence in the same genetic background used for immuno-EM, but, importantly, is not stained in an isogenic *△cdzAB* strain that lacks the type I secretion system. This new result corroborates our conclusion, based on cell fractionation and EM, that CdzC, and likely CdzD, are cell surface-exposed proteins. With regard to fimbriae or adhesins, we did not find any that are regulated in a manner similar to the *cdz* system, but the responsible structure could be constitutively produced or regulated in a different way.

*iii) The proline substitution mutants (Figure 7—figure supplement 1) are quickly glossed over, compared to the rest of the study. An alternative view is that the Pro mutant is unstable or fails to be exported. To exclude that and to substantiate the possibility that the zipper is important for killing activity as opposed to export/stability the authors should monitor secretion/surface exposure of this mutant variant. In absence of more controls the use of these mutants is not essential for the demonstration.*

We agree that these mutants were not characterized in depth and are not essential to any of the major conclusions or arguments in the paper, so we have removed them from the revised manuscript.

*iv) In their interspecies killing assays with Caulobacter crescentus Cdz producers, the authors have used two closely related α-proteobacteria. Have they used more distantly related members of this taxonomic group and can they comment on the phylogenetic distribution of potential Cdz prey? The Cdz mnemonic implies that only glycine zipper-like toxins are deployed by T1SS. Did the authors uncover other recognizable toxin-immunity genes linked to cdzAB related loci? Could this secretion mechanism be used to transfer nucleases, lipases or murein hydrolases into recipient cells?*

We have now tested another closely related species, *Asticcacaulis excentricus*, a member of the *Caulobacteraceae* family, and find that it is not killed. We also tested three additional, more distantly related α-proteobacteria, *Rhodobacter sphaeroides, Sinorhizobium meliloti,* and *Agrobacterium tumefaciens*, which are also not killed. These results indicate that CdzC/D is a relatively narrow spectrum toxin. We have added the new results to Figure 7—figure supplement 1, and updated the text (Results and Discussion) accordingly. As for whether CdzAB could transfer other types of toxins, we cannot, of course, say definitively, but our phylogenetic analysis does suggest that *cdzAB* homologs are commonly, though not exclusively, associated with bacteriocin-like peptides similar to CdzC/D. The proteins that are encoded near *cdzAB* in other organisms and that carry secretion leader sequences but are not similar to CdzC/D will have to be explored in future studies.